# Pillararene incorporated metal−organic frameworks for supramolecular recognition and selective separation

Yitao Wu[1,2,4], Meiqi Tang[1,4], Zeju Wang[1,2], Le Shi[1,2], Zhangyi Xiong[1,2], Zhijie Chen [1,2] ✉, Jonathan L. Sessler [3] ✉ & Feihe Huang [1,2] ✉

Crystalline frameworks containing incorporated flexible macrocycle units can afford new opportunities in molecular recognition and selective separation. However, such functionalized frameworks are difficult to prepare and challenging to characterize due to the flexible nature of macrocycles, which limits the development of macrocycle-based crystalline frameworks. Herein, we report the design and synthesis of a set of metal−organic frameworks (MOFs) containing pillar[5]arene units. The pillar[5]arene units were uniformly embedded in the periodic frameworks. Single crystal X-ray diffraction analysis revealed an interpenetrated network that appears to hinder the rotation of the pillar[5]arene repeating units in the frameworks, and it therefore resulted in the successful determination of the precise pillar[5]arene host structure in a MOF crystal. These MOFs can recognize paraquat and 1,2,4,5-tetracyanobenzene in solution and selectively remove trace pyridine from toluene with relative ease. The work presented here represents a critical step towards the synthesis of macrocycle-incorporated crystalline frameworks with well-defined structures and functional utility.

Host−guest chemistry plays a crucial role in nature and is closely related to the origin of life[1–3]. It has also allowed for critical advances in environmental science, drug delivery, chemical industry and sensing, among numerous other applications[4–7]. Macrocycles[8], such as crown ethers[9], cyclodextrins[10], calixarenes[11], cucurbiturils[12], and pillararenes[13], have been central to progress in host−guest chemistry and have permitted seminal advances in molecular recognition[14], separations[15], supramolecular materials development[16] and nanotechnology[17]. On the other hand, metal−organic frameworks (MOFs)[18–20]—a class of crystalline framework materials composed of organic struts and inorganic nodes—have been extensively explored for *inter alia* gas storage[21–23], water capture[24,25], and catalysis[26]. Not surprisingly, efforts have thus been made to incorporate macrocyclic subunits into

MOFs[27–29]. In principle, the resulting systems offer several prospective advantages, including: (1) enhanced regulation of the pore structures allowing for a fine-tuning of the molecular recognition and separation capabilities; (2) better accessibility to active recognition sites; (3) efficient diffusion of guest molecules; (4) clearer insights into structure−property relationships since each recognition site is isolated and amenable to independent study[27].

Unfortunately, crystalline frameworks incorporating well-defined flexible macrocyclic subunits remain challenging to prepare and difficult to characterize[30]. For example, although pillararene-based MOFs have been reported, their structural details remain recondite because rotations of the flexible pillararene subunits can lead to disorder within what are presumably overall periodic frameworks[31,\32].

[1]Stoddart Institute of Molecular Science, Department of Chemistry, Zhejiang University, Hangzhou 310058, P. R. China. [2]ZJU-Hangzhou Global Scientific and Technological Innovation Center-Hangzhou Zhijiang Silicone Chemicals Co., LTD Joint Lab, Zhejiang-Israel Joint Laboratory of Self-Assembling Functional Materials, ZJU-Hangzhou Global Scientific and Technological Innovation Center, Zhejiang University, Hangzhou 311215, P. R. China. [3]Department of Chemistry, The University of Texas at Austin, Austin, TX 78712-1224, USA. [4]These authors contributed equally: Yitao Wu, Meiqi Tang. ✉e-mail: zhijiechen@zju.edu.cn; sessler@cm.utexas.edu; fhuang@zju.edu.cn

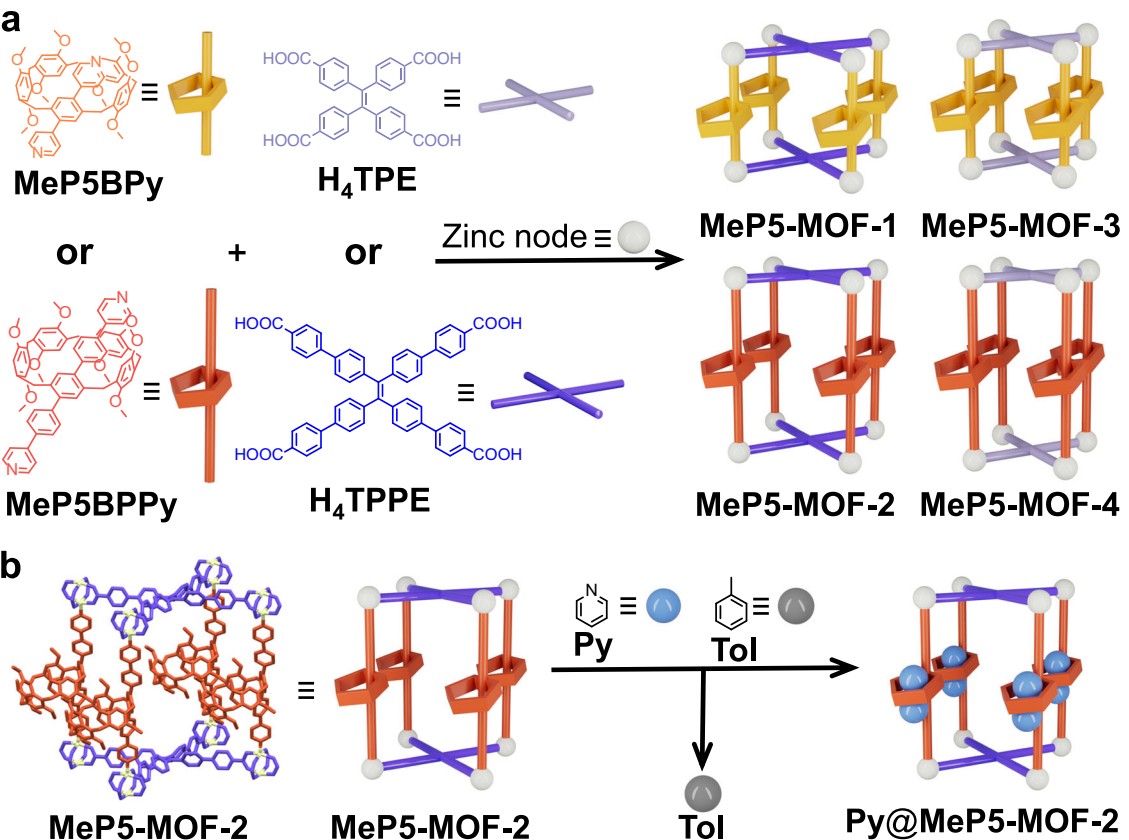

**Fig. 1 | Design and synthesis of pillar[5]arene-based MOFs. a** Cartoon representations and chemical structures of ligands and pillar[5]arene-based MOFs: **MeP5BPy, MeP5BPPy, H₄TPE, H₄TPPE**, zinc node, **MeP5-MOF-*n*** (*n* = 1, 2, 3, 4). **b** Schematic representations of the transformation from **MeP5-MOF-2** to **Py@MeP5-MOF-2** upon uptake of **Py** from a 90:10 *v/v* (87.3:12.7 mole percentage) **Tol/Py** mixture. **Py** = pyridine; **Tol** = toluene.

Herein, we report a set of pillar[5]arene-containing MOFs denoted as **MeP5-MOF-1, MeP5-MOF-2, MeP5-MOF-3**, and **MeP5-MOF-4** (Fig. 1a, b) via the so-called pillar-layer strategy[33–36]. In these systems, the pillar[5]arene-based **MeP5BPy** and **MeP5BPPy** subunits act as ligands for the zinc nodes and may be regarded as macrocycle-bearing struts, while the tetraphenylethylene (TPE) derivatives **H₄TPPE** and **H₄TPE** also complex the zinc centers and act as layers. As detailed below, single crystal X-ray diffraction (SCXRD) studies of the resulting pillar[5]arene-containing MOFs revealed network interpenetration in the case of **MeP5-MOF-2**. This interpenetration restricted the rotation of pillar[5]arene units within the frameworks and allowed the pillar[5]arene subunits to be resolved with atomic resolution. Two prototypical guests, paraquat (**PQT**) and 1,2,4,5-tetracyanobenzene (**TCN**), were tested as guests for **MeP5-MOF-1** and **MeP5-MOF-2**. Compared with **Model-MOF-1**, a MOF lacking an incorporated pillar[5]arene, **MeP5-MOF-1**, and **MeP5-MOF-2** displayed enhanced guest uptake. These two pillar[5]arene-bearing MOFs allowed for the effective removal of pyridine (**Py**) from toluene (**Tol**) with the toluene purity level up to 99.9% being reached rapidly. The observed selectivity is rationalized on the basis of single crystal structure analyses.

## Results and discussion
### Synthesis of pillar[5]arene-based struts and MOFs
Pillar[5]arene-containing struts **MeP5BPy** and **MeP5BPPy** were synthesized according to previous reports[37,38] (Supplementary Figs. 1–5). Single crystal structures confirmed that the lengths of these two struts, as judged by the N···N distances, are 11.41 and 19.71 Å, respectively (Supplementary Figs. 24–27). On this basis, we considered it likely that **MeP5BPy** and **MeP5BPPy** could be applied as struts in the "pillar-layer strategy" MOF preparation strategy pioneered by Kim and Hupp using

bipydine struts and zinc carboxylates[33,35,36] (Supplementary Figs. 6–17). In fact, by applying this strategy using **MeP5BPy** and **MeP5BPPy** in conjunction with **H₄TPPE** and **H₄TPE** it proved possible to prepare **MeP5-MOF-1, MeP5-MOF-2, MeP5-MOF-3**, and **MeP5-MOF-4** and obtain single crystals suitable for SCXRD analyses. Two model MOFs, **Model-MOF-1** and **MeModel-MOF-1**, lacking incorporated pillar[5]arene units were also prepared using 1,4-di(4-pyridyl) benzene (**PBPy**) and 4,4′-(2,5-dimethoxy-1,4-phenylene)dipyridine (**MePBPy**) as the struts.

### Structural determination of pillar[5]arene-based MOFs
Proton nuclear magnetic resonance (¹H NMR) spectroscopy was used to confirm the presence of the pillar[5]arene struts and TPE carboxylate layers. Taking **MeP5-MOF-1** as an example, signals corresponding to H$_{a-f}$ of **MeP5BPy** and H$_{g-j}$ of **H₄TPPE** were seen in the ¹H NMR spectrum (Supplementary Fig. 18). The peak integrations were found to match those expected for the structure as well as the mixture of ligands before complexation and after the **MeP5-MOF-1** was subject to digestion (treatment with DMSO-$d_6$/DCl (100:1 *v/v*) and subjecting to ultrasonication to give a transparent solution). Similar results were obtained for **Model-MOF-1, MeModel-MOF-1, MeP5-MOF-2, MeP5-MOF-3**, and **MeP5-MOF-4** (Supplementary Figs. 19–23). Further support for the assigned structures came from SCXRD analyses (Supplementary Data 1) as discussed below.

In the case of **MeP5-MOF-1**, the SCXRD analysis revealed a structure that is formulated as [Zn₂(**MeP5BPy**)(**TPPE**)], which consists of paddlewheel dinuclear Zn₂(COO)₄ secondary building units (SBUs) bound to carboxylate and pyridine linkers (Supplementary Figs. 28, 29). Each **TPPE** is coordinated to four Zn₂ nodes to form two-dimensional layers. These layers are joined together by **MeP5BPy** to

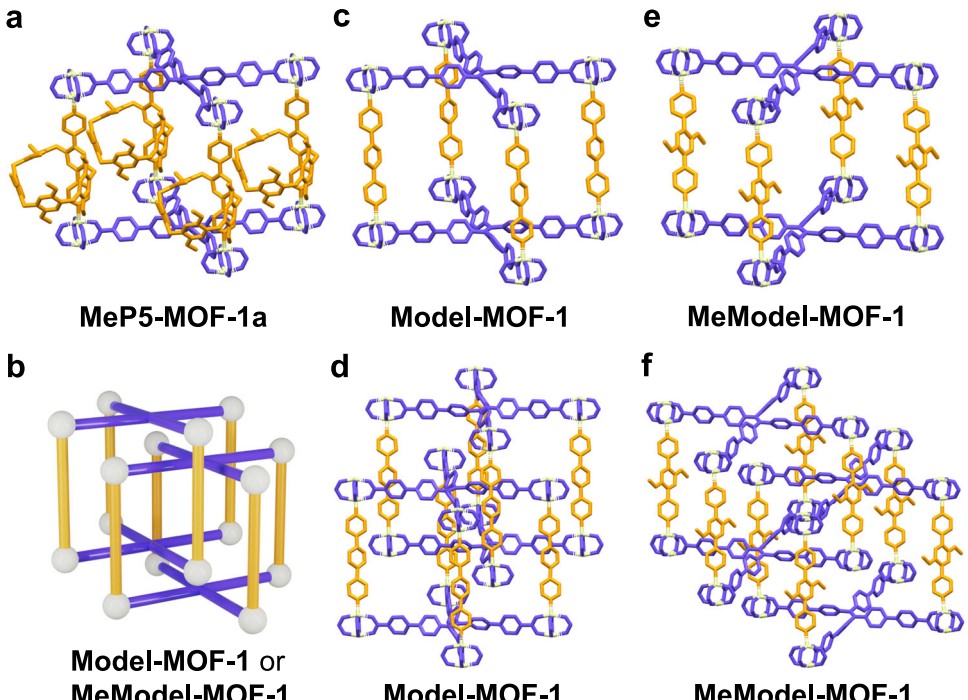

**Fig. 2 | Structures of MeP5-MOF-1, Model-MOF-1, and MeModel-MOF-1.**
**a** Capped-stick representation of structural model **MeP5-MOF-1a** of **MeP5-MOF-1**.
Here the backbone of **MeP5-MOF-1** was determined by SCXRD, while the pillar[5]
arene units are the results of structural model. **b** Cartoon representation of **Model-**

**MOF-1** or **MeModel-MOF-1**. **c–f** Capped-stick representations of single crystal
structures of **Model-MOF-1** and **MeModel-MOF-1**. The TPE ligands are blue, pil-
lared struts are orange, zinc nodes are white, hydrogen atoms and solvent mole-
cules have been omitted for clarity.

form a non-interpenetrated **fsc**-type framework (Fig. 2a). The experi-
mental powder X-ray diffraction (PXRD) pattern for **MeP5-MOF-1** was
similar to that simulated from the SCXRD structure, confirming the
phase purity and crystallinity (Supplementary Fig. 66). It is worth
noting that only the non-pillar[5]arene backbone is observed in the
crystal structure of **MeP5-MOF-1**. Presumably, this reflects the fact that
dynamics of the pillar[5]arene units preclude their being located pre-
cisely within the overall framework. Given this absence of direct
observation, a structural model, **MeP5-MOF-1a**, with the pillar[5]arene
units shown was constructed (Fig. 2a and Supplementary Figs. 30, 31).
The experimental SCXRD data for **MeP5-MOF-1** was used as the
starting point for constructing this model. The PXRD pattern simu-
lated from the calculated structure of **MeP5-MOF-1a** proved consistent
with that of an as-synthesized **MeP5-MOF-1** sample (Supplemen-
tary Fig. 32).

Single crystals of the control MOFs, **Model-MOF-1** and **MeModel-
MOF-1**, lacking incorporated pillar[5]arenes were also obtained. The
resulting structures revealed that **Model-MOF-1** and **MeModel-MOF-1**
are comprised of [Zn₂(**PBPy**)(**TPPE**)](DMF)₂ and [Zn₂(**MePBPy**)
(**TPPE**)](DMA)₂ entities. As above, ¹H NMR spectral analysis confirmed
the presence of the expected constituent ligands in both **Model-MOF-
1** and **MeModel-MOF-1** (Supplementary Figs. 19, 20). The PXRD pat-
terns of the as-synthesized samples also provided support for the
purity and crystallinity of these materials (Supplementary Figs. 67, 68).
Of interest is that the crystal structures of **Model-MOF-1** and
**MeModel-MOF-1** revealed two-fold interpenetrated polymeric frame-
works. This stands in contrast to what was seen for **MeP5-MOF-1**, a
framework characterized by a non-interpenetrated structure as noted
above. This disparity provides support for the suggestion that the
presence or absence of incorporated pillar[5]arene units can control
whether or not an interpenetrated MOF structure is obtained (Fig. 2b–f
and Supplementary Figs. 33–36).

The observation of interpenetrated structures in the case of
**Model-MOF-1** and **MeModel-MOF-1** provided a motivation to analyze

in detail the other MOFs prepared in the context of the present study.
A further incentive was to explore whether it would be possible to
obtain a structure wherein the incorporated pillar[5]arenes could be
located unambiguously. Doing so would address a recognized need
since in most reported receptor-incorporated MOFs the flexible mac-
rocycles cannot be located, presumably on account rotation-based
disorder as inferred in the case of **MeP5-MOF-1**[27,31,\32,\39,\40].

Single crystals of **MeP5-MOF-2** were obtained via a solvo-thermal
procedure, wherein strut **MeP5BPPy** was combined with 2 equivalent
of **H₄TPPE** and 2 equivalent of Zn(NO₃)₂·6H₂O in DMF (Supplementary
Fig. 9). Acetic acid was added as a modulator. The crystal structure of
**MeP5-MOF-2** (Fig. 3a) revealed that it contains extended pillared struts
and is composed of [Zn₂(**MeP5BPPy**)(**TPPE**)] entities. As true for
**MeP5-MOF-1**, **MeP5-MOF-2** possesses a "pillar-layer" structure. How-
ever, in contrast to **MeP5-MOF-1**, **MeP5-MOF-2** contains two-fold
interpenetrated networks. Moreover, the pillar[5]arene units in **MeP5-
MOF-2** could be resolved by SCXRD (Fig. 3b and Supplementary
Figs. 37–42). Here, in order to distinguish the dynamics of the pillar[5]
arene units in **MeP5-MOF-1** and **MeP5-MOF-2**, the number of pillar[5]
arene units in a 1 nm³ volume element within the single crystal struc-
tures were calculated for both systems[40]. Compared with what was
seen for single crystal structures of the struts **MeP5BPy** and
**MeP5BPPy** (0.80 and 0.76 pillar[5]arene units per 1 nm³ volume ele-
ment, respectively; Supplementary Table 17), **MeP5-MOF-1** contains
only 0.16 pillar[5]arene units per 1 nm³ volume element while the
corresponding value of **MeP5-MOF-2** is 0.23. This low density reflects
empty space around the pillar[5]arene and makes both **MeP5-MOF-1**
and **MeP5-MOF-2** dynamic with the latter system less so (Supple-
mentary Figs. 56–59). Calculations regarding the ability of the pillar[5]
arene units to rotate in **MeP5-MOF-1** and **MeP5-MOF-2** were also car-
ried out. The results revealed that the pillar[5]arene units in **MeP5-
MOF-1** could rotate 45° along the struts at a minimum potential energy
point after optimization. The corresponding value was 10° in **MeP5-
MOF-2**, which leads us to suggest that the pillar[5]arene units in **MeP5-**

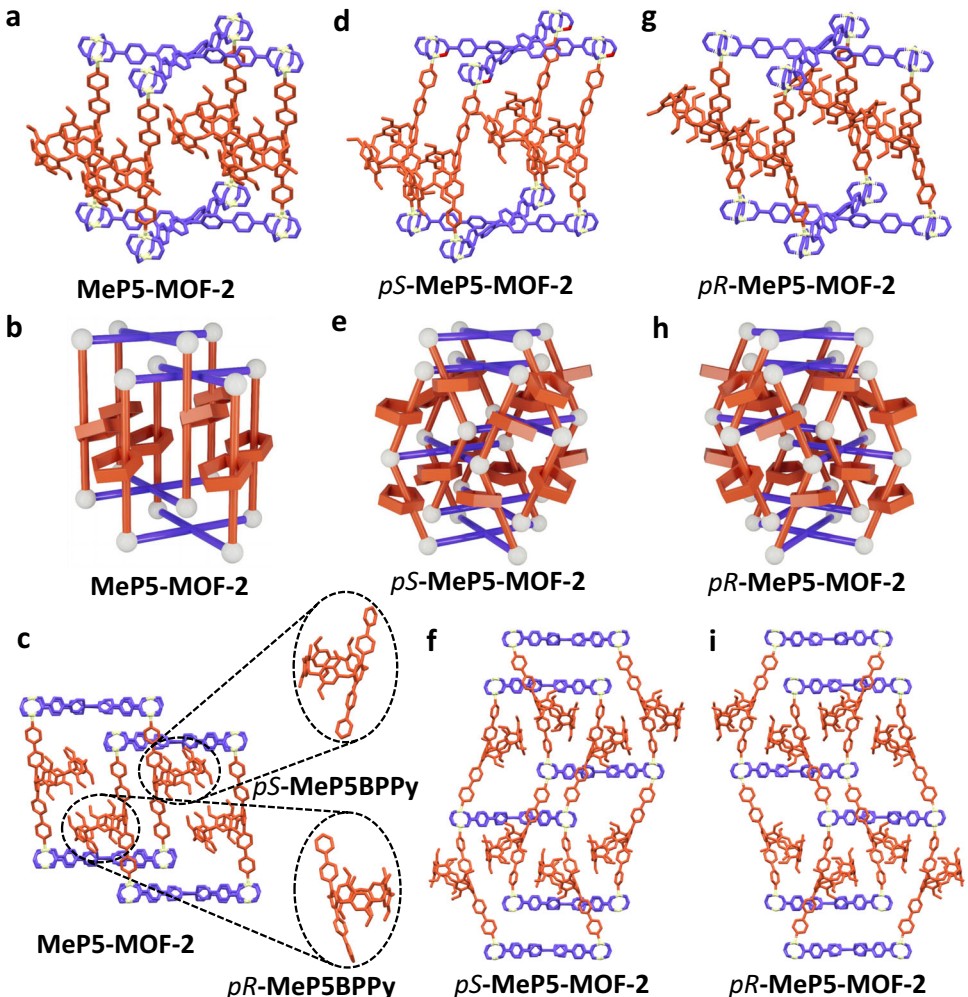

**Fig. 3 | Structures of MeP5-MOF-2, *pS*-MeP5-MOF-2, and *pR*-MeP5-MOF-2.**
**a**–**i** Single crystal structures shown in capped-stick form and cartoon representations of **MeP5-MOF-2, *pS*-MeP5-MOF-2**, and *pR*-**MeP5-MOF-2**. The TPE ligands are blue, pillared struts are red, zinc nodes are white, and hydrogen atoms have been omitted for clarity.

**MOF-1** are more flexible than those in **MeP5-MOF-2** (Supplementary Figs. 60, 61).

SCXRD data of **MeP5-MOF-1** were collected at 105 K in an effort to ascertain whether a lower temperature would limit the dynamics of the pillar[5]arene units within the framework. From the single crystal structure of **MeP5-MOF-1**, the pillar[5]arene units could still not be visually characterized even at 105 K (Supplementary Figs. 62, 63). Therefore, we ascribe our ability to observe the pillar[5]arene units in **MeP5-MOF-2** to the fact that they occupy the internal voids of the frameworks in a pairwise stacked manner, which presumably limits their motion. Each pair consists of **MeP5BPPy** units in their respective *pS* and *pR* conformations (Fig. 3c). The experimental PXRD pattern also provided support for the purity and crystallinity of the **MeP5-MOF-2** sample (Supplementary Fig. 69).

In order to explore the influence of planar chirality of the pillar[5]arene units in stabilizing the observed rigid **MeP5-MOF-2** framework, pre-resolution of *racemic*-**MeP5BPPy** was performed. Two enantiomers *pS*-**MeP5BPPy** and *pR*-**MeP5BPPy** (Supplementary Figs. 45–47) were separated and used to fabricate the corresponding chiral MOFs, *pS*-**MeP5-MOF-2** and *pR*-**MeP5-MOF-2** (Supplementary Figs. 48–51). Single crystals of both species were obtained. The Flack parameters for *pS*-**MeP5-MOF-2** and *pR*-**MeP5-MOF-2** were found to be 0.28(2) and 0.26(3). These values lead us to conclude that the absolute structures of these species could not be fully determined, probably due to the partial conformational interconversion between the *pS*/*pR*-**MeP5** congeners during the preparation of *pS*/*pR*-**MeP5-MOF-2** (Supplementary Tables 9, 10)[41,42]. As true for *racemic*-**MeP5-MOF-2**, both *pS*-**MeP5-MOF-2** and *pR*-**MeP5-MOF-2** consisted of two-fold interpenetrated frameworks. However, the pillar[5]arene struts are oriented toward the layers; in contrast, those of *racemic*-**MeP5-MOF-2** are perpendicular (Fig. 3d–i). Compared to *racemic*-**MeP5-MOF-2**, the simulated PXRD patterns of *pS*- and *pR*-**MeP5-MOF-2** from the SCXRD analysis also revealed changes, as would be expected given the differences in the internal spatial arrangement of the constituent pillar[5]arene units (Supplementary Fig. 70).

When the TPE ligand was changed from $H_4TPPE$ to $H_4TPE$, pillar[5]arene-containing MOFs **MeP5-MOF-3** and **MeP5-MOF-4** were obtained from **MeP5BPy** and **MeP5BPPy**, respectively. As true for **MeP5-MOF-1** (and other macrocycle-incorporating systems), the pillar[5]arene units on the struts of **MeP5-MOF-3** and **MeP5-MOF-4** still could not be located because of disorder (Supplementary Figs. 52–55). Nevertheless, the experimental PXRD patterns confirmed the phase purity and crystallinity of these two systems (Supplementary Figs. 71, 72). Taken in concert, the studies of **MeP5-MOF-*n*** (*n* = 1–4) provide support for the intuitively appealing conclusion that frameworks containing struts with incorporated pillar[5]arene units need to limit the dynamics of the flexible macrocycle moiety sufficiently if the macrocycles are to be visualized effectively by SCXRD.

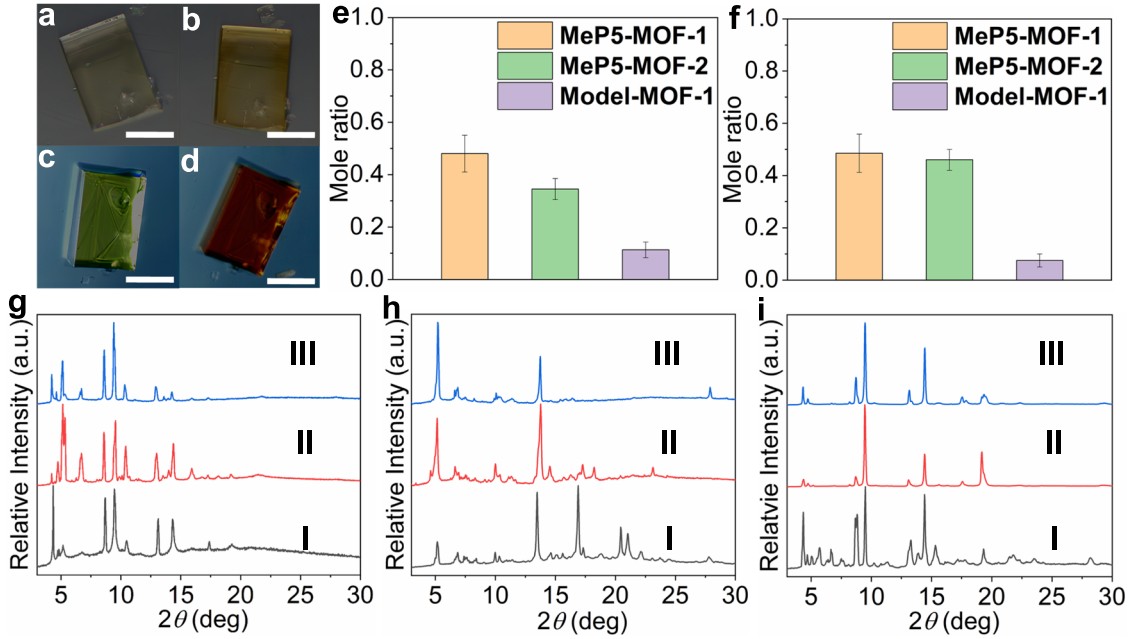

**Fig. 4 | Supramolecular recognition studies of MeP5-MOF-1 and MeP5-MOF-2 with PQT and TCN.** Optical microscopy images of **MeP5-MOF-1: a** before uptake of **PQT**; **b** after uptake of **PQT**; **c** before uptake of **TCN**; **d** after uptake of **TCN**. Scale bars, 200 μm. The mole ratios of **PQT** (**e**) and **TCN** (**f**) to struts in **MeP5-MOF-1**, **MeP5-MOF-2**, and **Model-MOF-1**, as inferred from $^1$H NMR spectral studies of these MOFs after guest uptake in acetone. **g**–**i** PXRD patterns of single crystalline samples of **MeP5-MOF-1, MeP5-MOF-2**, and **Model-MOF-1: I** before guest uptake; **II** after uptake of **PQT**; **III** after uptake of **TCN**. Source data of **g**–**i** are provided as a Source data file.

## Supramolecular recognition of pillar[5]arene-based MOFs for PQT and TCN

The clear structural differences between **MeP5-MOF-1** and **MeP5-MOF-2** (open and interpenetrated) led us to explore their molecular recognition features. Two substrates, namely **PQT** and **TCN**, that are known to be bound by pillar[5]arenes[43,44], were chosen as guests for these studies. Fluorescence spectroscopic titration experiments in acetone were performed to determine the association constants and binding stoichiometries between the pillar[5]arenes and guests (Supplementary Figs. 73–88). On this basis, association constants ($K_a$) of $55 \pm 1\,M^{-1}$, $41 \pm 1\,M^{-1}$, $53 \pm 1\,M^{-1}$, and $36 \pm 1\,M^{-1}$ were calculated for **PQT@MeP5BPy**, **PQT@MeP5BPPy**, **TCN@MeP5BPy**, and **TCN@MeP5BPPy**, respectively. A 1:1 binding stoichiometry was inferred in all four cases.

The solid-state recognition features were also explored. Initially, crystals of **MeP5-MOF-1** exhibited a pale-yellow color but changed to dark yellow after uptake of **PQT** (Fig. 4a, b). Presumably, this darkening reflects charge transfer interactions between the pillar[5]arene units and the guests. **MeP5-MOF-2** also underwent a color change upon exposure to **PQT**. In contrast, almost no color change was seen for the control, **Model-MOF-1**, under otherwise identical conditions (Supplementary Figs. 89–91). The extent of uptake was determined by calculating the mole ratios of the guests to the struts using the integrated intensities of the corresponding $^1$H NMR signals. The $^1$H NMR spectra of **MeP5-MOF-1, MeP5-MOF-2**, and **Model-MOF-1** after uptake of **PQT** and digestion in DMSO-$d_6$/DCl (100:1 $v/v$) revealed that the mole ratios of guests to struts were about 0.48, 0.35, and 0.11 (Fig. 4e and Supplementary Figs. 97–99) for these three MOFs, respectively.

In the case of **TCN**, crystalline **MeP5-MOF-1** and **MeP5-MOF-2** changed from yellow to red (Fig. 4c, d). This observation was taken as evidence of uptake and the formation of complexes characterized by strong charge transfer interactions between the pillar[5]arene units and the **TCN** guests within the frameworks. In contrast, little change in color was observed in the case of **Model-MOF-1** (Supplementary Figs. 92–94). Based on analyses analogous to those carried out in the

case of **PQT**, the **TCN** uptake by **MeP5-MOF-1, MeP5-MOF-2**, and **Model-MOF-1** was about 0.49, 0.46, and 0.08, respectively (Fig. 4f and Supplementary Figs. 100–102). PXRD patterns of **MeP5-MOF-1, MeP5-MOF-2**, and **Model-MOF-1** after uptake of guests revealed evidence for a change in structure while maintaining crystallinity (Fig. 4g–i). Overall, **MeP5-MOF-1** displayed a somewhat higher level of uptake than **MeP5-MOF-2**, which in turn displayed more effective guest uptake than **Model-MOF-1** lacking incorporated pillar[5]arene units. We thus conclude that the pillar[5]arene units incorporated into MOFs play an active role in supramolecular recognition and that guest uptake is not dictated by simple diffusion.

## Separation of Tol/Py mixtures using pillar[5]arene-based MOFs

The potential utility of pillar[5]arene-containing MOFs for separations was tested using **Tol** and **Py**. **Tol** is one of the most important raw materials in the chemical industry[45]. However, **Tol** is typically contaminated with trace quantities of **Py**[46]. It is necessary to remove residual **Py** from **Tol** to obtain high-quality **Tol**. This is a challenging separation compounded by the fact that **Tol** and **Py** possess similar boiling points (b.p.) (**Tol**: 110.60 °C; **Py**: 115.50 °C; Supplementary Table 18). Moreover, an azeotrope (b.p. 110.20 °C) forms under conditions of distillation[47]. To explore whether the present MOFs could provide a useful alternative, an initial solid–liquid experiment was carried out by soaking crystalline **MeP5-MOF-1** in single component **Tol** or **Py** for two minutes. The crystals were collected by filtration. The relative uptake of **Tol** or **Py** was measured by calculating the mole ratios of **Tol** or **Py** relative to the struts using NMR spectroscopy. This analysis revealed that **MeP5-MOF-1** could capture 0.6 equivalent of **Tol** and 6.7 equivalents of **Py** *per* pillar[5]arene unit (Supplementary Figs. 109, 110). After solid–liquid adsorption, NMR analysis revealed a 90.8% selectivity for **Py** obtained from a 100 μL of 90:10 $v/v$ (87.3:12.7 mole percentage) **Tol/Py** mixture using ~20 mg of crystalline **MeP5-MOF-1** (Supplementary Fig. 111). A separate quantitative analysis was also performed by heating the samples to release the adsorbed guests and monitoring the volatiles using gas chromatography (GC). This

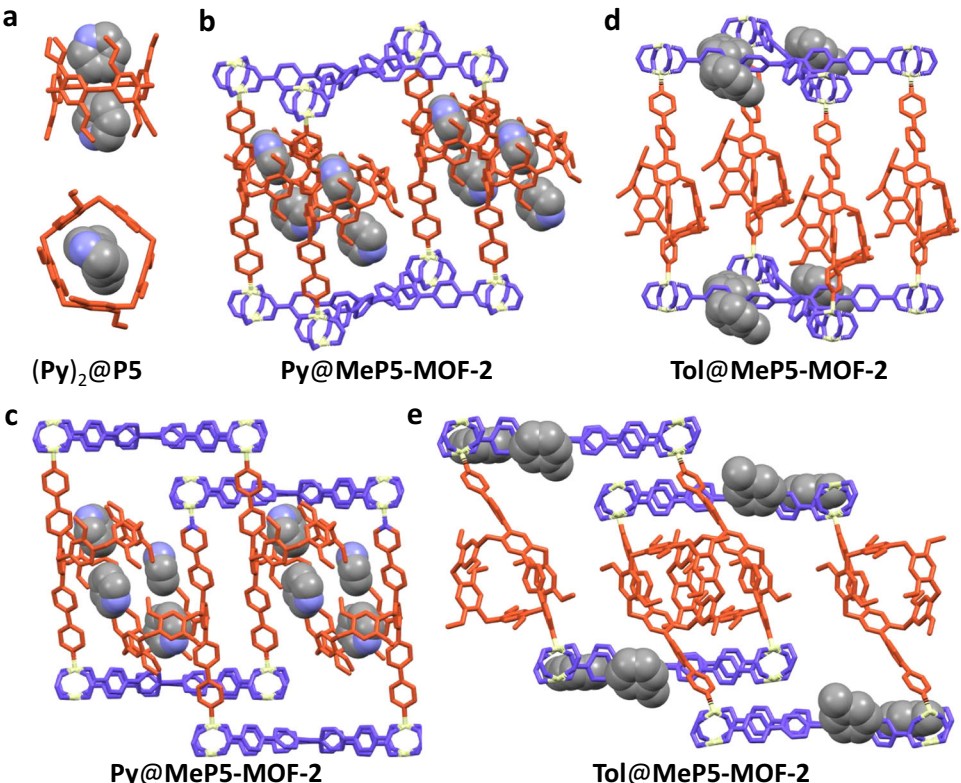

**Fig. 5 | Structures of (Py)₂@P5, Py@MeP5-MOF-2, and Tol@MeP5-MOF-2.** Illustrated structures of **a** (Py)₂@P5, **b**, **c** Py@MeP5-MOF-2, and **d**, **e** Tol@MeP5-MOF-2. The occupancy of **Tol** in MeP5-MOF-2 is 0.5. Here (**a**), (**d**), and (**e**) are views of single crystal structures, while the **Py** molecules in (**b**) and (**c**) are models based on the single crystal structure of **MeP5-MOF-2** and (**Py**)₂@**P5**. TPE ligands are blue, pillared struts are red, zinc nodes are white. Hydrogen atoms have been omitted for clarity. **Py** and **Tol** are shown in the spacefilling form and color coded by element.

study revealed that (1) the samples adsorbed **Py** with 90.3% selectivity, a finding consistent with the NMR spectroscopic analysis (Supplementary Fig. 112) and that (2) the mole percentage of **Tol** in the above mixture increased to 96.9% from the initial 87.3% value when ~20 mg samples were used (Supplementary Fig. 113). The mole percentage of **Tol** further increased to 99.9% after the product of the initial separation was treated with another ~20 mg of fresh samples (Supplementary Fig. 114). NMR spectroscopic analyses confirmed that roughly 77% of the adsorbed guests in **MeP5-MOF-1** could be removed by washing with acetone five times in succession (Supplementary Fig. 115). An analogous study was carried out starting with a 99:1 *v/v* **Tol/Py** mixture. In this case, the GC analysis revealed that the mole percentage of **Tol** increased from 98.7% to 99.9% (Supplementary Fig. 116). PXRD analyses showed that **MeP5-MOF-1** changed its arrangement after guest uptake; presumably, this change reflects the dynamic nature of the framework (Supplementary Fig. 117).

MeP5-MOF-2 was also tested for its ability to separate **Tol/Py** mixtures. In this case, initial NMR spectral tests confirmed that **MeP5-MOF-2** can accommodate 2.8 equivalents of **Tol** and 2.6 equivalents of **Py** *per* pillar[5]arene unit in single component experiments (Supplementary Figs. 118, 119). A ca. 90.0% selectivity was seen for **Py** adsorbed from a 100 µL of a 90:10 *v/v* **Tol/Py** mixture using ~20 mg sample of crystalline **MeP5-MOF-2** (Supplementary Fig. 120). GC analyses indicated that the corresponding selectivity of **Py** was 89.5% which matched what was inferred from the NMR studies (Supplementary Fig. 121). The mole percentage of **Tol** in the resulting mixture increased from 87.3% to 90.5% under these conditions (Supplementary Fig. 122). Roughly 87% of the guests in **MeP5-MOF-2** could be removed upon washing with acetone five times (Supplementary Fig. 123). A PXRD analysis confirmed that, as true for **MeP5-MOF-1**, **MeP5-MOF-2** also

changed its arrangement upon uptake of guests (Supplementary Fig. 124).

An effort was made to obtain diffraction grade single crystals of **Py@MeP5-MOF-2**. Unfortunately, no suitable crystals could be obtained, perhaps as the result of the disordered nature of the **Py** molecules within the frameworks. Given this, single crystals of **Py** in 1,4-diethoxypillar[5]arene (**P5**) were grown by dissolving powdered **P5** in **Py** and allowing to evaporate at room temperature for about one week. The resulting crystal structure revealed that one pillar[5]arene molecule can accommodate two **Py** molecules within its cavity (Fig. 5a). Based on the metric parameters, **Py** binding is driven by [C−H⋯O] and [C−H⋯π] interactions ([C⋯O] distances (Å), [H⋯O] distances (Å) and [C−H⋯O] angles (deg) of [C−H⋯O] hydrogen bonds: 3.42, 2.54, 154.88; 3.42, 2.54, 154.88. [C−H⋯π] distances (Å) and angles (deg): 3.00, 158.34; 3.07, 158.76. Supplementary Figs. 125–127).

To predict the preferred location of **Py** molecules within the present MOF frameworks, we carried out sorption module of location simulations based on the single crystal structures of **MeP5-MOF-2** and (**Py**)₂@**P5** (Supplementary Figs. 128–130). These simulations revealed that the adsorbed **Py** molecules are retained in the pillar[5]arene cavities (Fig. 5b, c). Single crystals of **Tol@MeP5-MOF-2** were obtained by immersing **MeP5-MOF-2** in a solution consisting of DMF and **Tol** (5:1 *v/v*) for a day. An ensuing SCXRD analysis revealed that as compared with **MeP5-MOF-2**, the unit cell of **Tol@MeP5-MOF-2** is relatively constrained. Moreover, the **Tol** molecules are found within the voids between the **TPPE** layers rather than in the pillar[5]arene cavities (Supplementary Figs. 132–134). This stands in contrast to what is seen for **Py@P5**. We rationalize this difference in terms of the smaller size of the **Py** molecules which makes them more likely to be trapped in the pillar[5]arene cavities. We thus suggest that the pillar[5]arene units

incorporated into **MeP5-MOF-2** endows the system with an ability to capture **Py** selectively from **Py/Tol** mixtures through a macrocycle-dependent recognition process and that this effect is enhanced by confinement within a framework (Fig. 5d, e).

The experimental PXRD patterns of crystalline **Py@MeP5-MOF-2** and **Tol@MeP5-MOF-2** do not quite match the corresponding simulated patterns. While not a proof, this disparity could reflect dynamics in the MOF structure (Supplementary Figs. 131 and 135). In contrast, no obvious selectivity of contacting with **Py** or **Tol** guests was seen in the case of **Model-MOF-1** (Supplementary Figs. 136–140). A solid–vapor experiment also confirmed that 1,4-dimethoxypillar[5]arene (**MeP5**) lacks selectivity in adsorbing **Py** from **Tol/Py** mixtures (Supplementary Figs. 141–145). This result is taken as evidence that confinement within a suitable MOF framework improves the selectivity for **Py**[48]. We thus conclude that a judicious choice of receptor (e.g., pillar[5]arene) and MOF framework (e.g., **MeP5-MOF-2**) allows for the specific removal of **Py** from **Py/Tol** mixtures.

We further investigated the efficiency of both **MeP5-MOF-1** and **MeP5-MOF-2** under bulk conditions (~200 mg). Both **MeP5-MOF-1** and **MeP5-MOF-2** gave similar results with samples (~20 mg for each) tested as adsorbents (Supplementary Figs. 148–153). The recyclability of the separation process was also tested. Host–guest studies of **MeP5-MOF-1** and **MeP5-MOF-2** revealed that some guest molecules were still trapped in **MeP5-MOF-1** and **MeP5-MOF-2** after each cycle according to $^1$H NMR spectroscopic analyses. The corresponding PXRD analyses revealed a partial loss in crystallinity in both **MeP5-MOF-1** and **MeP5-MOF-2** after each cycle of guest uptake (Supplementary Figs. 154, 155). We also investigated the environmental tolerance of **MeP5-MOF-1** and **MeP5-MOF-2** under various treatment conditions. The porosity of **MeP5-MOF-1** and **MeP5-MOF-2** was studied through $CO_2$ and $N_2$ adsorption/desorption measurements (Supplementary Figs. 156–161). Experimental $CO_2$ adsorption/desorption isotherms at 195 K measuring the porosity of activated **MeP5-MOF-1** and **MeP5-MOF-2** revealed apparent Brunauer-Emmett-Teller (BET) surface areas of 160 m²/g and 190 m²/g, respectively. These MOFs are non-porous to $N_2$ as revealed by $N_2$ sorption experiments at 77 K. The difference between $CO_2$ and $N_2$ is ascribed to the fact that these MOFs showed relative strong affinity for $CO_2$ compared to $N_2$ according to another related report[49]. TGA studies of **MeP5-MOF-1** and **MeP5-MOF-2** were performed to investigate their thermal stability. The resultant TGA curves revealed that **MeP5-MOF-1** had only a 3.6% weight loss before around 150 °C, which was assigned to solvent loss, and began to decompose at around 350 °C while samples of **MeP5-MOF-2** had a 3.9% weight loss before around 150 °C and began to decompose at around 400 °C (Supplementary Figs. 162–165). After treatment with some specific solvents, the PXRD patterns revealed that **MeP5-MOF-1** and **MeP5-MOF-2** still maintained their crystallinity (Supplementary Figs. 166, 167). PXRD patterns corresponding to wet and vacuum treatment were also analysed and proved consistent with the notion that these MOFs were dynamic and would lose their partial crystallinity under some conditions (Supplementary Figs. 168–171).

In conclusion, we have designed and synthesized a set of MOFs incorporating pillar[5]arene motifs. By comparing the structures of these MOFs determined by SCXRD methods we conclude that network interpenetration in the case of **MeP5-MOF-2** plays a crucial role in allowing the pillar[5]arene units in the frameworks to be resolved. In other words, we suggest that hindering the rotation of the pillar[5]arene repeating units eliminates their crystallographic disorder within the frameworks. Both **MeP5-MOF-1** and **MeP5-MOF-2** were found to adsorb **PQT** and **TCN** well. In contrast, the pillar[5]arene-free control MOF, **Model-MOF-1**, proved relatively ineffective. In addition, both **MeP5-MOF-1** and **MeP5-MOF-2** could be used to achieve the separation of **Py** from **Tol** efficiently and with relative ease. The observation of clear structure–property relationships paves the way for the rational construction of functional framework materials embedded with supramolecular moieties for use in specific recognition and separation application.

## Methods

### Single crystal growth
Single crystals of **MeP5BPPy** were grown by dissolving 5.00 mg of dry **MeP5BPPy** powder in chloroform, heating until all the powder was dissolved and allowing to evaporate about one week. Single crystals of (**Py**)$_2$@**P5** were grown by placing 5.00 mg of dry **P5** powder in a small vial, adding 1 mL of **Py**, heating until all the powder was dissolved, and allowing to evaporate at room temperature about one week. Single crystals of **Tol@MeP5-MOF-2** were obtained by immersing **MeP5-MOF-2** in a solution consisting of DMF and **Tol** (5:1 $v/v$) for a day.

### General procedure for preparing single crystals of MOFs
Taking **MeP5-MOF-1** as an example: A DMF suspension (1.5 mL) of **MeP5BPy** (8.50 mg, 10.0 μmol), **H₄TPPE** (8.10 mg, 10.0 μmol), and Zn(NO₃)₂·6H₂O (5.97 mg, 20.0 μmol) was prepared in a small vial. This suspension was sonicated two minutes and then passed through a syringe filter to give a transparent solution, which was sealed, heated at a constant rate of 1 °C min⁻¹ to 90 °C, kept at that temperature for 48 h and cooled to room temperature at a constant cooling rate of 0.2 °C min⁻¹. Transparent flaxen-colored single crystals of **MeP5-MOF-1** suitable for SCXRD were obtained and followed by immersion in 12 mL of acetone for 3 days, with the solvent topped off twice daily.

## Data availability
The authors declare that all other data supporting the findings of this study are available in the Article and its Supplementary Information. Crystallographic data of the structures reported in this Article are available from the Cambridge Crystallographic Data Centre (CCDC) with the following codes: Single crystal X-ray diffraction data for **MeP5BPPy** (CIF), CCDC number 2211420; **MeP5-MOF-1** (CIF), CCDC number 2211421; (DMF)$_2$@**Model-MOF-1** (CIF), CCDC number 2217237; (DMA)$_2$@**MeModel-MOF-1** (CIF), CCDC number 2213797; **MeP5-MOF-2** (CIF), CCDC number 2211422; DMF@**MeP5-MOF-2** (CIF), CCDC number 2211423; (DMF)$_3$@**MeP5-MOF-2** (CIF), CCDC number 2217239; **Model-MOF-2** (CIF), CCDC number 2211425; $pS$-**MeP5-MOF-2** (CIF), CCDC number 2217240; $pR$-**MeP5-MOF-2** (CIF), CCDC number 2217241; **MeP5-MOF-3** (CIF), CCDC number 2211424; **MeP5-MOF-4** (CIF), CCDC number 2217410; (**Py**)$_2$@**P5** (CIF), CCDC number 2216441; **Tol@MeP5-MOF-2** (CIF), CCDC number 2217238; **MeP5-MOF-1-105K** (CIF), CCDC number 2267727; **MeP5-MOF-1-G** (CIF), CCDC number 2267730. These data can be obtained free of charge from The Cambridge Crystallographic Data Centre via www.ccdc.cam.ac.uk/structures. Cif and checkcif files of all the crystallographic data in this work are provided as a Supplementary Data 1 file. Source data are provided with this paper.

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

## Acknowledgements

This work was supported by the National Key Research and Development Program of China (2021YFA0910100 to F.H.), the National Natural Science Foundation of China (22035006 to F.H., 22201247 to Z.C.), the Zhejiang Provincial Natural Science Foundation of China (LD21B020001 to F.H.), and the Starry Night Science Fund of Zhejiang University Shanghai Institute for Advanced Study (SN-ZJUSIAS-006 to F.H.), and a Leading Innovation Team grant from Department of Science and Technology of Zhejiang Province (2022R01005 to F.H.). The work in Austin was supported by the Robert A. Welch Foundation (F-0018 to J.L.S.). Z.C. thanks Zhejiang University for startup funding. We thank Yaqin Liu, Ling He, Mingxin Yu, and Jiyong Liu from the Chemistry Instrumentation Center of Zhejiang University for technical support. We thank Yaer Zhu of the Analysis Center of the Agrobiology and Environmental Sciences & Institute of Agrobiology and Environmental Sciences of Zhejiang University for the GC experiments. We thank Wendi Chen from Shiyanjia Lab (www.shiyanjia.com) for the SCXRD analysis. We thank eceshi (www.eceshi.com) for the NMR tests.

## Author contributions

Y.W., Z.C., and F.H. conceived and designed the experiments with advice from J.L.S. Y.W., and M.T. performed the experiments. Y.W., L.S., Z.X., and Z.C. analyzed the data. Y.W. wrote the paper. Y.W., M.T., Z.W., L.S., Z.X., Z.C., J.L.S., and F.H. revised the paper. All authors contributed to the data analysis and discussion.

## Competing interests

The authors declare no competing interests.
