## [Peer Review File · Nature Communications]

Pillararene incorporated metal–organic frameworks for supramolecular recognition and selective separationREVIEWER COMMENTS

Reviewer #1 (Remarks to the Author):

This work describes the synthesis of metal-organic frameworks (MOFs) that incorporate pillar[5]arene (PA) struts, utilizing 4,4'-bipyridinyl ligands as pillars. The pillars are employed in a "pillar-layer strategy" that links Zn nodes and 4-connected planar carboxylate layers to produce extended framework structures. Notably, the MOFs exhibit selective separation of toluene/pyridine through molecular recognition, specifically in MeP5-MOF-1 and MeP5-MOF-2. The authors suggest that the use of solid-liquid adsorption with these MOFs could result in more efficient separations compared to conventional methods.

The approach utilized in this study is a widely used method for synthesizing pillar-layer structure MOFs, involving the assembly of zinc carboxylate paddlewheel SBUs with linear bipyridine ligands as pillaring struts. Although this strategy has been employed previously to incorporate a macrocyclic moiety into the pillar ligand of MOF (as described in *J. Am. Chem. Soc.* 2014, 136, 20, 7403-7409), it is worth noting that such pillar-layer structures are still relatively uncommon among PA-based MOFs that have been reported to date.

Although there have been recent reports on the use of nonporous adaptive crystals of PA for selective separation of Tol/Py (Ref. S14-S16), to the best of my knowledge, such applications have not yet been reported in MOFs. Therefore, this work demonstrates a significant level of novelty, and I recommend that it be considered for publication in *Nature Communications* if the authors thoroughly address the critical issues outlined below.

1. Sentences in lines 50-54 may be potentially misleading to the reader. It is difficult for me to agree with the assertion that all frameworks containing macrocyclic subunits are challenging to characterize. It is necessary to clarify that this statement pertains only to flexible macrocycles.
2. The authors need to provide further clarification on why the rotation of the PA subunit hinders the determination of a single crystal structure. The references cited by the authors (Ref. 27, 31, 32, 39, and 40) do not provide clear evidence to support this claim. For instance, it would be helpful to know if it is also challenging to identify the SC-XRD of a crystal packed solely with pure PA. (S24-25)
3. In the flexible CE-based MOF, macrocycle dynamics were previously studied through 2H SSNMR (as reported in *J. Am. Chem. Soc.* 2014, 136, 20, 7403-7409). Is there any data available that can directly demonstrate the rotation of the PA ring in MeP5-MOFs, as presented in the aforementioned study?
4. It is widely recognized that SCXRD at low temperatures can limit the mobility of subunits within crystals and provide more precise structural information (*Chem. Soc. Rev.*, 2004, 33, 490-500). Therefore, the authors must supply SCXRD data collected below 193 K for each framework.
5. The cartoon presented in Fig 1a might impede an intuitive comprehension of these MOF structures and introduce ambiguity to the nomenclature. It is crucial to explicitly specify which moiety each MOF contains and the corresponding structural characteristics.
6. For "After numerous attempts" on line 145, this is an unnecessary sentence.
7. For "The ability ... limits their motion" on lines 152-154, this is a reasonable claim, but authors also need direct experimental data to prove it.
8. The authors need to offer further background information on why the crystal's absolute structure can be justified if the crystal's SCXRD has a Flack constant below 0.3.
9. Regarding the statement "Take in concert, ... effectively by SCXRD" in lines 176-180, the authors require concrete evidence to support their assertions. For instance, would a de-interpenetrated MeP5-MOF-2 yield the same outcome as 1, 3, and 4? Additionally, if an extra moiety capable of restricting the pore space in MeP5-MOF-1, 3, and 4 is integrated into the linker, would it enable the identification of the PA ring of the pillar?
10. The photographs in Figures 4a and b are not distinctly displaying the color differences among the crystals because of the varying background brightness levels in the two images.
11. In addition to the fluorescence spectra of the PA unit, the authors should also demonstrate whether guest adsorption alters the fluorescence spectra of each MOF.
12. The author needs to clarify the toluene/pyridine separation condition of other studies to be compared. For example, in Ref. S14, 5.00 mg of adsorbent was added to 10 mL of 100:1 v/v Tol/Py mixture.
13. Regarding the statement "We thus conclude ... Py/Tol mixture separation capability" on lines

299-301, the author needs to be cautious about this claim. The experimental conditions for the Tol/Py separation performance experiment in this work differ from those of the studies used for comparison by the authors.

14. The author should provide data comparing the performance of their MOF with currently available commercial Tol/Py separation processes.

15. The authors should provide additional data on the Tol/Py separation experiments, including information on the efficiency of the MOF adsorbents in bulk conditions, as well as their recyclability and stability in the Tol/Py separation process.

16. The authors should provide experimental data on the porosity of MeP5-MOFs, ideally through gas adsorption/desorption isotherm measurements.

Reviewer #2 (Remarks to the Author):

In this work, the authors present the synthesis of a series of MeP5-MOF-n single crystals based on pillar[5]arene-based ligands through a "pillar-layer" strategy. The introduction of macrocycle units into MOF frameworks exhibited intriguing guest recognition and separation properties. The efforts are impressive to grow and pick the single crystals and refine the SCXRD structure of complicated macrocycle-based MOFs with soft and disordered pillar[5]arene units. The current work is suitable for publication in Nature Communication after following minor revisions:

1. It is recommended that the authors review the illustration figures for better clarity. In the current state, it is hard to distinguish different linkers of the same series (like MeP5BPY/MeP5BPPY and H4TPE/H4TPPE). I recommend using different colors for the same series of linkers (for example red for MeP5BPY and orange for MeP5BPPY, blue for H4TPE and purple for H4TPPE).
2. In Figure S27, some C-C bonds on the benzene rings of ligands seem strange, showing square-shaped bonding. This figure should be revised.
3. SCXRD indicated rigid framework backbone structures of the MeP5-MOF-n series. Is there any information about the permanent porosity of these MOFs from gas sorption analysis? How is the stability of these MOFs towards thermal treatment and ambient moisture?
4. MeP5-MOF-1 showed higher guest uptake than MeP5-MOF-2, what's the possible reason?
5. For Py and Tol separation, simulation and SCXRD refinement indicated that Py was most likely captured by the cavities of pillar[5]arene rings, while Tol was instead adsorbed between TPPE layers. Such distinct adsorption behaviors should be discussed in more detail from a chemical perspective to better understand the underlying guest separation mechanism.
6. What's the guest sorption and separation properties of MeP5-MOF-3 and MeP5-MOF-4? No related properties were examined in the current manuscript.
7. The determination of flexible groups in the structure through SCXRD may be more accurate at lower temperatures. Can SCXRD at lower temperatures (e.g., 80 K) be tested?
8. The "Mole Ratio" in Fig. 4 should be added with error bar.
9. The format of references should be noted, such as "macrocycle-based" in reference [16] and "clathrochelate-based" in reference [46].

Reviewer #3 (Remarks to the Author):

The authors reported the synthesis of pillarene-based MOFs and their use in host-guest recognition and separation of small molecules. Steric effect has been effective in hindering the rotation of linker pillars that resulted in the successful resolution of macrocyclic pillarenes within the MOF structure, which has been a challenge in previous related studies. The structural advance provides unequivocal evidence about the presence and location of macrocycles within the framework. The pillarene-MOFs have shown improved and selective adsorption towards guests, and in one case being applied towards the separation of pyridine/toluene. Structural characterization was done thoroughly and thoughtfully. The authors however didn't address the stability of the resulting MOFs, which raises substantial questions about their value towards host-guest binding and separation.

The manuscript is very well written overall, though I recommend the authors to address the

following comments before its acceptance:

1. Porosity. For the interpenetrated MOF vs. the non-interpenetrated ones, are there significant differences in terms of surface area and pore size? Have the authors run BET studies of those MOFs?
2. The PXRD pattern of MOFs changed significantly before and after exposure to different guests. The authors attributed the changes to the dynamics within the MOF. This is quite handwaving. Can authors give more insight into such changes? This raises the question about the stability of the MOF towards solvents, vacuum, temperature etc. Are the host-guest responses reversible, i.e., can the PXRD be reverted after removal of guests? If not, what is the nature of the solid-state changes? It is reasonable to believe that the MOFs are assembled through the coordination between pyridyl groups of the pillars and the Zn metal centers, which is weak and may relate to the intrinsic instability of such MOFs. The authors are also suggested to run DSC to probe the thermal stability of the crystalline phases.
3. On a related note, as stated, better separation was achieved by repeated treatment of the Py/Tol mixture with fresh batches of MOF. Does the need of using fresh MOF also relate to the instability of the MOFs? The significance of their utility for separation will be greatly limited since it is of no practicality if the adsorbent can't be recycled.

Reviewer #4 (Remarks to the Author):

In this manuscript, the authors reported a set of metal-organic frameworks (MOFs) containing incorporated pillar[5]arene units and used them in molecular recognition and selective separation. Since numerous macrocycle-incorporated MOFs cannot precisely locate the macrocycle moieties in the frameworks due to the rotation of macrocycles' repeat units, the structure resolution of such MOFs is always a challenge. In this work, the authors altered the sizes of pillar[5]arene struts and obtained the atomically precise structures of pillar[5]arene-based MOFs characterized by single crystal X-ray diffraction. They found that the interpenetrated network limited the rotation of the pillar[5]arene repeat units in the frameworks. It is an interesting phenomenon, which is helpful for constructing macrocycle-incorporated crystalline frameworks in other cases. Their MOFs can recognize paraquat and 1,2,4,5-tetracyanobenzene in solution and selectively remove trace pyridine from toluene.

This manuscript is well written, and clearly illustrates the structure-property relationships with solid evidence. Therefore, I recommend this work to be accepted for publication after considering the following minor comments.

1. Page 6, "After numerous attempts, single crystals of MeP5-MOF-2 were obtained utilizing a solvo-thermal procedure, strut MeP5BPPy was combined with H4TPPE and Zn(NO₃)₂·6H₂O." Here, the authors should describe the conditions of single crystal growth with more details.
2. Page 7, "Single crystals of both species were obtained. The Flack parameters for pS-MeP5-MOF-2 and pR-MeP5-MOF-2 were found to be 0.28(2) and 0.26(3). These values are less than 0.30, allowing their absolute structures to be determined with confidence." Here, the authors should cite the corresponding references about Flack parameters.
3. In this manuscript, the authors used TPE ligands as the layers to construct their pillar[5]arene-based MOFs. Were there any reasons to use the TPE ligands?

RESPONSE TO REVIEWERS' COMMENTS

We very appreciate the valuable comments made by the four reviewers. In aggregate, they were very helpful and allowed us to improve our manuscript. We recognize the wisdom of the reviewers and have revised the manuscript and supplementary information as requested by the reviewers. The corrections and improvements are listed below in point by point fashion in response to the reviewers' comments.

Reviewer #1's comments:

Although there have been recent reports on the use of nonporous adaptive crystals of PA for selective separation of **Tol/Py** (Ref. S14–S16), to the best of my knowledge, such applications have not yet been reported in MOFs. Therefore, this work demonstrates a significant level of novelty, and I recommend that it be considered for publication in *Nature Communications* if the authors thoroughly address the critical issues outlined below.

Response: We thank the reviewer for reading our manuscript so carefully and for having summarized the significance of our paper clearly. We also thank the reviewer for the positive comments. Our point-by-point response is given below:

1. Reply to the first comment made by Reviewer 1 "Sentences in lines 50–54 may be potentially misleading to the reader. It is difficult for me to agree with the assertion that all frameworks containing macrocyclic subunits are challenging to characterize. It is necessary to clarify that this statement pertains only to flexible macrocycles."

Many thanks. The corresponding corrections have been made in the manuscript. The first two sentences in the second paragraph were changed to: "Unfortunately, crystalline frameworks incorporating well-defined flexible macrocyclic subunits remain challenging to prepare and difficult to characterize³⁰. For example, although pillararene-based MOFs have been reported, their structural details remain recondite because rotations of the flexible pillararene subunits can lead to disorder within what are presumably overall periodic frameworks^{31–32}." Here "flexible" was added to describe the macrocyclic subunits.

2. Reply to the second comment made by Reviewer 1 "The authors need to provide further clarification on why the rotation of the PA subunit hinders the determination of a

single crystal structure. The references cited by the authors (Ref. 27, 31, 32, 39, and 40) do not provide clear evidence to support this claim. For instance, it would be helpful to know if it is also challenging to identify the SC-XRD of a crystal packed solely with pure PA. (S24-25)”.

Here, in order to distinguish the dynamics of pillar[5]arene units in **MeP5-MOF-1** and **MeP5-MOF-2** frameworks, the contents of pillar[5]arene units in 1 nm³ volume elements in the single crystal structures were calculated (*Mater. Today Chem.* **25**, 100973 (2022)).

The SCXRD of **MeP5BPy** and **MeP5BBPy** can be simply identified since multiple noncovalent interactions such as [C–H···O], [C–H···N], [C–H···π] and [π···π] interactions exist between adjacent pillar[5]arene molecules. In contrast to the pillar[5]arene units in the frameworks, pillar[5]arene molecules are packed together in the “pure pillar[5]arene” state while they are isolated and flexible in the frameworks. Therefore, compared with the single crystal structures of pillared struts **MeP5BPy** and **MeP5BBPy** (0.80 and 0.76 pillar[5]arene unit in 1 nm³ volume elements, respectively; the CCDC number of **MeP5BPy** is 2096367 according to a previous report: *J. Am. Chem. Soc.* **143**, 11976–11981 (2021)), **MeP5-MOF-1** has only 0.16 pillar[5]arene unit in 1 nm³ volume elements while the corresponding value of **MeP5-MOF-2** is 0.23. This low density reflects empty space around the pillar[5]arene and makes both **MeP5-MOF-1** and **MeP5-MOF-2** dynamic with the latter system less so, resulting the pillar[5]arene units in **MeP5-MOF-1** harder to be resolved.

Supplementary Fig. 25 Capped-stick representation of the single crystal structure of **MeP5BPy** showing the packing arrangement. Carbon atoms are grey, oxygen atoms are red, and nitrogen atoms are blue. Hydrogen atoms are omitted for clarity. The CCDC number of **MeP5BPy** is 2096367 per a previous report^{S2}.

Supplementary Fig. 27 Capped-stick representation of the single crystal structure of **MeP5BPPy** showing the packing arrangement. Carbon atoms are grey, oxygen atoms are red, and nitrogen atoms are blue. Hydrogen atoms are omitted for clarity.

Supplementary Table 17 Density calculation of pillar[5]arene units

Substance	Number of pillar[5]arene units per unit cell	Volume of unit cell (nm ³)	Number of pillar[5]arene units per cubic nanometer (nm ⁻³)
MeP5BPPy	4	4.99	0.80
MeP5-MOF-1	1	6.08	0.16
MeP5BPPy	4	5.25	0.76
MeP5-MOF-2	2	8.59	0.23

Supplementary Fig. 56 Capped-stick representation of the single crystal structure of **MeP5BPY** in a unit cell. Carbon atoms are grey, oxygen atoms are red, and nitrogen atoms are blue. Hydrogen atoms are omitted for clarity.

Supplementary Fig. 57 Capped-stick representation of the calculated structure for **MeP5-MOF-1a** based on the backbone of **MeP5-MOF-1** in a unit cell. Carbon atoms are grey, oxygen atoms are red, nitrogen atoms are blue, and zinc atoms are dark blue. Hydrogen atoms are omitted for clarity.

Supplementary Fig. 58 Capped-stick representation of the single crystal structure of **MeP5BPPy** in a unit cell. Carbon atoms are grey, oxygen atoms are red, and nitrogen atoms are blue. Hydrogen atoms are omitted for clarity.

Supplementary Fig. 59 Capped-stick representation of the single crystal structure of **MeP5-MOF-2**. Carbon atoms are grey, oxygen atoms are red, nitrogen atoms are blue, and zinc atoms are dark blue. Hydrogen atoms are omitted for clarity.

Calculations on the rotation ability of pillar[5]arene units in **MeP5-MOF-1** and **MeP5-MOF-2** were further conducted. From the calculated structures based on single crystal structures, the pillar[5]arene units in **MeP5-MOF-1** are able to rotate 45° at a minimum potential energy point after optimization while the corresponding value of **MeP5-MOF-2** is only 10°, which suggested that the pillar[5]arene units in **MeP5-MOF-1** are more flexible compared with those of **MeP5-MOF-2**. Based on the single crystal structures and the calculations, we concluded that the rotation of the pillar[5]arene subunit hinders the determination of a whole single crystal structure of **MeP5-MOF-1**.

Supplementary Fig. 60 Capped-stick representation of the calculated structure **MeP5-MOF-1a** based on the backbone of **MeP5-MOF-1**. Here, the pillar[5]arene unit on the strut is optimized to rotate 45° at a minimum potential energy point. Carbon atoms are grey, oxygen atoms are red, nitrogen atoms are blue, and zinc atoms are dark blue. Hydrogen atoms are omitted for clarity, 1 a.u. = 627.51 kcal/mol.

Supplementary Fig. 61 Capped-stick representation of the calculated structure based on the single crystal structure of **MeP5-MOF-2**. Here, the pillar[5]arene unit on the strut is optimized to rotate 10° at a minimum potential energy point. Carbon atoms are grey,

oxygen atoms are red, nitrogen atoms are blue, and zinc atoms are dark blue. Hydrogen atoms are omitted for clarity, 1 a.u. = 627.51 kcal/mol.

Intrinsic dynamics of the pillar[5]arene units impose great challenges on solving the crystal structure based on the diffraction data. It is well known that the *p*-dimethoxy phenyl rings have very low rotation barrier and thus bring rotational disorder to the pillared struts (*Nat. Commun.* **14**, 590 (2023)). Pillar[5]arene units have planar chirality and their racemic mixture are used for the MOF synthesis. This is on top of the inherent flexibility of the pillar[5]arenes. These disorders make **MeP5-MOF-1** a great example of “robust dynamics”, in which each pillar[5]arene unit enjoys a high freedom of dynamics and the pillar[5]arene units are anchored to a periodic robust structure. The pillar[5]arene disorders make the electron density very diffused in SCXRD. Most of the carbon atoms in the pillared strut units show high atomic displacement parameters, while the other carbon atoms in the pillar[5]arenes are impossible to be located.

Since the pillar[5]arene units attached to the struts show rotational disorder around the struts, planar chirality (racemic disorder), and inherent flexibility, and thus make the electron density very diffused. Diffractions at high theta range were very weak, making the R_{int} higher than a satisfactory value. Therefore, only positions of all the atoms in the SBUs, the layers and the pillar[5]arene-lacked struts could be unambiguously determined after several restraints were applied.

The complete MOF structure model of **MeP5-MOF-1** is constructed by attaching pillar[5]arene units to the framework backbone solved by SCXRD. The single crystal structure of the strut **MeP5BPpy** provides a good model fragment to accurately describe the pillar[5]arene structure in an extended framework. Thus, by grafting the pillar[5]arene units onto the linear struts, we construct a definite structure model of **MeP5-MOF-1a**.

Supplementary Fig. 30 Capped-stick representation of the calculated structure **MeP5-MOF-1a** based on the backbone of **MeP5-MOF-1**. The pillar[5]arene units on the struts are shown. Carbon atoms are grey, oxygen atoms are red, nitrogen atoms are blue, and zinc atoms are dark blue. Hydrogen atoms are omitted for clarity.

Supplementary Fig. 31 Polyhedral representation of the calculated structure **MeP5-MOF-1a** along the c axis (a) and the b axis (b). The elementary cell is marked with an orange cuboid ($a = 16.45 \text{ \AA}$, $b = 20.20 \text{ \AA}$, $c = 18.31 \text{ \AA}$). Carbon atoms are grey, hydrogen atoms are white, oxygen atoms are red, nitrogen atoms are blue, and zinc atoms are dark blue.

Supplementary Fig. 32 PXRD patterns: **I**, simulated from the single crystal structure of **MeP5-MOF-1**; **II**, simulated from the calculated structure of **MeP5-MOF-1a**; **III**, **MeP5-MOF-1** from a single crystal sample.

3. Reply to the third comment made by Reviewer 1 “In the flexible CE-based MOF, macrocycle dynamics were previously studied through ^2H SSNMR (as reported in *J. Am. Chem. Soc.* 2014, 136, 20, 7403–7409). Is there any data available that can directly demonstrate the rotation of the PA ring in MeP5-MOFs, as presented in the aforementioned study?”

Many thanks for the valuable advice. In this study, we cannot investigate the pillar[5]arene dynamics in the frameworks through ^2H SSNMR due to the lack of deuterium sites on the pillar[5]arene struts **MeP5BPy** and **MeP5BPPy**. VT PXRD can

reflect the dynamics of **MeP5-MOFs** but is unable to distinguish the dynamics of the frameworks and pillar[5]arenes.

Here, the dynamics and rotation calculations of pillar[5]arene units in **MeP5-MOF-1** and **MeP5-MOF-2** were performed and shown in our response to Question 2 of Reviewer #1. These results revealed the dynamics of pillar[5]arene units in the frameworks and also suggested that the pillar[5]arene units in **MeP5-MOF-1** are more flexible compared with those of **MeP5-MOF-2**.

Moreover, we also synthesized another pillar[5]arene-based MOF denoted as **MeP5-MOF-6** with flexible crown ether units as the “layer” (this work will be published in another paper). Single crystals of **MeP5-MOF-6** were obtained and SCXRD data were collected. The single crystal structure revealed consider crystallographic disorder, presumably because of the flexible nature of the crown ether units and the pillar[5]arenes in the frameworks. Even though the crystal structure revealed a two-fold interpenetrated network, the strong rotation of the pillar[5]arene units can still be observed in the resolved single crystal structure, albeit with disorder. The preparation, single crystal structure and SCXRD data of **MeP5-MOF-6** are summarized below:

Additional Fig. 1 Preparation of **MeP5-MOF-6**.

Additional Fig. 2 Capped-stick representation of the single crystal structure of **MeP5-MOF-6**. Here, the rotation of pillar[5]arene along the strut in **MeP5-MOF-6** is crystallographically confirmed. Carbon atoms are grey, oxygen atoms are red, nitrogen atoms are blue, and zinc atoms are dark blue. Hydrogen atoms are omitted for clarity.

Additional Fig. 3 Capped-stick representation of the single crystal structure of **MeP5-MOF-6** in a unit cell. Carbon atoms are grey, oxygen atoms are red, nitrogen atoms are blue, and zinc atoms are dark blue. Hydrogen atoms are omitted for clarity.

Additional Table 1 Experimental SCXRD data of **MeP5-MOF-6**

Formula	[Zn ₂ (MeP5BPy)(B18C6)]
Empirical formula	C ₁₁₃ H ₉₆ N ₂ O ₂₂ Zn ₂
Formula weight	1964.65
Temperature/K	193
Crystal system	monoclinic
Space group	P 2 ₁ / m
a /Å	11.7141(10)
b /Å	25.1385(18)
c /Å	26.7305(18)
α /°	90
β /°	101.979(2)
γ /°	90
Volume/Å ³	7700.0(10)
Z	2
ρ_{calc} (g/cm ³)	0.847
μ /mm ⁻¹	0.359
F (000)	2048
Crystal size/mm ³	0.13 × 0.12 × 0.1
Radiation	MoK α (λ = 0.71073)

2 θ range for data collection/ $^{\circ}$	3.512 to 50.698
Index ranges	$-14 \leq h \leq 13, -30 \leq k \leq 30, -32 \leq l \leq 32$
Reflections collected	58513
Independent reflections	14391 [$R_{\text{int}} = 0.0644, R_{\text{sigma}} = 0.0626$]
Data/restraints/parameters	14391/1944/885
Goodness-of-fit on F^2	1.134
Final R indexes [$I \geq 2\sigma(I)$]	$R_1 = 0.1086, wR_2 = 0.3014$
Final R indexes [all data]	$R_1 = 0.1503, wR_2 = 0.3310$
Largest difference peak/hole/ $e \text{ \AA}^{-3}$	0.73/-0.61
CCDC-number	unpublished

4. Reply to the fourth comment made by Reviewer 1 “It is widely recognized that SCXRD at low temperatures can limit the mobility of subunits within crystals and provide more precise structural information (Chem. Soc. Rev., 2004, 33, 490-500). Therefore, the authors must supply SCXRD data collected below 193 K for each framework.”

Many thanks. Here, due to the lower quality of single crystals and more disorder in the crystal structures of **MeP5-MOF-3** and **MeP5-MOF-4**, **MeP5-MOF-1** was selected as a representative system for low temperature measurements. The SCXRD data of **MeP5-MOF-1** were collected at 105 K with the goal of ascertaining whether this relatively low temperature would limit the dynamics of the pillar[5]arene units in the frameworks. Unfortunately, in the single crystal structure of **MeP5-MOF-1**, the pillar[5]arene units could still not be visually characterized. This finding was taken as evidence for the motion-limited visualization of pillar[5]arene units in the **MeP5-MOF-2** frameworks. The single crystal structure and SCXRD data of **MeP5-MOF-1** collected at 105 K are shown below:

Fig. 17 Capped-stick representation of the single crystal structure of **MeP5-MOF-1** measured at 105 K. The pillar[5]arene units on the pillar struts are disordered and still

unsolved. Carbon atoms are grey, oxygen atoms are red, nitrogen atoms are blue, and zinc atoms are dark blue. Hydrogen atoms are omitted for clarity.

Fig. 18 Polyhedral representation of the single crystal structure of **MeP5-MOF-1** measured at 105 K along the *c* axis (a) and the *b* axis (b). The elementary cell is marked with an orange cuboid ($a = 16.18 \text{ \AA}$, $b = 20.37 \text{ \AA}$, $c = 18.24 \text{ \AA}$). The pillar[5]arene units on the pillar struts are disordered and still unsolved. Carbon atoms are grey, hydrogen atoms are white, oxygen atoms are red, nitrogen atoms are blue, and zinc atoms are dark blue.

Supplementary Table 15 Experimental SCXRD data for **MeP5-MOF-1** collected at 105 K

Formulas	[Zn ₂ (MeP5BPy)(TPPE)]
Empirical formula	C ₃₅ H ₂₂ NO ₄ Zn
Formula weight	585.9
Temperature/K	105
Crystal system	orthorhombic
Space group	Pmmm
a /Å	16.181(12)
b /Å	18.240(10)
c /Å	20.371(15)
α /°	90
β /°	90
γ /°	90
Volume/Å ³	6012(7)
Z	2
ρ_{calc} (g/cm ³)	0.324
μ /mm ⁻¹	0.236
F (000)	602
Crystal size/mm ³	0.13 × 0.12 × 0.1
Radiation	GaK α ($\lambda = 1.34139$)

2θ range for data collection/ $^{\circ}$	4.752 to 107.812
Index ranges	$-19 \leq h \leq 19, -21 \leq k \leq 21, -24 \leq l \leq 24$
Reflections collected	100053
Independent reflections	6087 [$R_{\text{int}} = 0.0749, R_{\text{sigma}} = 0.0315$]
Data/restraints/parameters	6087/66/151
Goodness-of-fit on F^2	0.957
Final R indexes [$I \geq 2\sigma(I)$]	$R_1 = 0.1298, wR_2 = 0.3036$
Final R indexes [all data]	$R_1 = 0.1402, wR_2 = 0.3135$
Largest difference peak/hole/ $e \text{ \AA}^{-3}$	1.28/-0.91
CCDC-number	2267727

5. Reply to the fifth comment made by Reviewer 1 “The cartoon presented in Fig 1a might impede an intuitive comprehension of these MOF structures and introduce ambiguity to the nomenclature. It is crucial to explicitly specify which moiety each MOF contains and the corresponding structural characteristics.”

Many thanks. The corresponding corrections have been made to Fig 1a in the manuscript as shown below:

Fig. 1 | Design and synthesis of pillar[5]arene-based MOFs. a, Cartoon representations and chemical structures of ligands and pillar[5]arene-based MOFs: **MeP5BPpy**, **MeP5BPPy**, **H₄TPE**, **H₄TPPE**, zinc node, **MeP5-MOF-*n*** ($n = 1, 2, 3, 4$). **b,** Schematic representations of the transformation from **MeP5-MOF-2** to **Py@MeP5-MOF-2** upon uptake of **Py** from a 90:10 v/v (87.3:12.7 mole percentage) **Tol/Py** mixture. **Py** = pyridine;

Tol = toluene.

6. Reply to the sixth comment made by Reviewer 1 “For “After numerous attempts” on line 145, this is an unnecessary sentence.”

Many thanks. The corresponding corrections have been made as shown below: “Single crystals of **MeP5-MOF-2** were obtained *via* a solvo-thermal procedure, wherein strut **MeP5BPPy** was combined with 2 equiv of **H4TPPE** and 2 equiv of $\text{Zn}(\text{NO}_3)_2 \cdot 6\text{H}_2\text{O}$ in DMF (Supplementary Fig. 9)” Here, “After numerous attempts” was deleted in this sentence.

7. Reply to the seventh comment made by Reviewer 1 “For ‘The ability to observe the pillar[5]arene units is ascribed to the fact that they occupy the internal voids of the frameworks in a pairwise stacked manner, which presumably limits their motion.’ on lines 152–154, this is a reasonable claim, but authors also need direct experimental data to prove it.”

Many thanks. The dynamics and rotation calculations of pillar[5]arene units in **MeP5-MOF-1** and **MeP5-MOF-2** frameworks were performed. The results are shown in our response to Question 2 of Reviewer #1. The density calculation of pillar[5]arene units revealed that limiting the motion of pillar[5]arene units in the frameworks was beneficial for the resolution of the pillar[5]arene units.

The single crystal structure and SCXRD data of **MeP5-MOF-6** were additionally provided in our response to Question 3 of Reviewer #1 which also indicated that even interpenetration limits motion of the pillar[5]arene units in the frameworks and that the pillar[5]arene units could be resolved by SCXRD. However, there are still considerable disorder in the framework due to the flexible nature of crown ether moieties and pillar[5]arene units. SCXRD data for **MeP5-MOF-1** were also collected at 105 K to investigate whether lower temperatures could limit the dynamics of the pillar[5]arene units in the frameworks. From the single crystal structure of **MeP5-MOF-1**, the pillar[5]arene units could still not be visually characterized even at 105 K. Therefore, these results revealed that the key to observing the pillar[5]arene units in this case is to limit the space for motion of the pillar[5]arene units within the frameworks.

8. Reply to the eighth comment made by Reviewer 1 “The authors need to offer further background information on why the crystal’s absolute structure can be justified if the crystal’s SCXRD has a Flack constant below 0.3.”

Many thanks for the valuable advice. The Flack parameter x , and its standard uncertainty u are explained in terms of the inversion-distinguishing power $x(u)$, which can evaluate absolute-structures and absolute-configurations (*Acta Cryst.* **A39**, 876–881 (1983)). Usually, the absolute structure of a crystal can be considered validated when the Flack parameter is below 0.1 with an enantiopure-sufficient inversion-distinguishing power. The corresponding references are cited in the manuscript as Ref. 41 and 42: *J. Appl. Cryst.* **33**, 1143–1148 (2000); *Angew. Chem. Int. Ed.* **25**, 11809–11813 (2021).

In the manuscript, the Flack parameter of *pS*-**MeP5-MOF-2** and *pR*-**MeP5-MOF-2** are 0.28(2) and 0.26(3), respectively. The standard uncertainty u of them is smaller than 0.04,

which means the inversion-distinguishing power is strong. However, the Flack parameter is not sufficiently robust to determine the absolute structures, probably due to the partial conformation conversion of the *pS/pR*-**MeP5** forms during the preparation of *pS/pR*-**MeP5-MOF-2** even we used pure chiral pillar[5]arene struts to construct the MOFs. According to the literature, the *p*-dimethoxy phenyl rings have very low rotation barrier and thus bring rotational disorder to the pillared struts which, in turn, induces the presumed conformation inversion (*Nat. Commun.* **14**, 590 (2023)). We apologize for the misunderstanding of the absolute configuration determination in the original manuscript, and thank this reviewer for helping us avoid an error of misinterpretation. The corresponding corrections have been made in the manuscript and are provided below: “The Flack parameters for *pS*-**MeP5-MOF-2** and *pR*-**MeP5-MOF-2** were found to be 0.28(2) and 0.26(3), respectively. These values lead us to conclude that the absolute structures of these species could not be fully determined, probably due to the partial conformational interconversion between the *pS/pR*-**MeP5** congeners during the preparation of *pS/pR*-**MeP5-MOF-2** (Supplementary Tables 9–10)^{41–42}.”

9. Reply to the ninth comment made by Reviewer 1 “Regarding the statement ‘Taken in concert, the studies of *MeP5-MOF-n* ($n = 1-4$) provide support for the intuitively appealing conclusion that frameworks containing struts with incorporated pillar[5]arene units need to be sufficiently open to allow formation of interpenetrated structures if the macrocycles are to be visualized effectively by SCXRD’ in lines 176-180, the authors require concrete evidence to support their assertions. For instance, would a de-interpenetrated *MeP5-MOF-2* yield the same outcome as 1, 3, and 4? Additionally, if an extra moiety capable of restricting the pore space in *MeP5-MOF-1*, 3, and 4 is integrated into the linker, would it enable the identification of the PA ring of the pillar?”

Many thanks. We agree with the reviewer’s comment that interpenetration is not an absolute condition to resolve the structure of pillar[5]arene units in the frameworks. The statement has been revised in the manuscript: “Taken in concert, the studies of **MeP5-MOF-n** ($n = 1-4$) provide support for the intuitively appealing conclusion that frameworks containing struts with incorporated pillar[5]arene units need to limit the dynamics of flexible macrocycle moiety sufficiently if the macrocycles are to be visualized effectively by SCXRD.” The dynamics and rotation calculations of pillar[5]arene units were discussed in our response to Questions 2, 3 and 7 of Reviewer #1, and we found that interpenetration was actually a method to limit the motion of pillar[5]arene units in the frameworks but not the only one. Single crystal structure and SCXRD data for **MeP5-MOF-6** are provided in our response to Question 3 of Reviewer #1, which indicated even though interpenetration limits the dynamics of the pillar[5]arene units in the frameworks, there is still considerable disorder in the frameworks due to the flexible nature of the crown ether moieties and pillar[5]arene units within the frameworks. We tried to obtain a de-interpenetrated **MeP5-MOF-2** through regulating the reaction conditions but failed to achieve success.

Additionally, we also tried to use an extra moiety to locate the pillar[5]arene units in the MOFs. Considering the quality of single crystals, we used **MeP5-MOF-1** as an example. Pillar[5]arene has been shown to capture TCN in its cavity (*J. Am. Chem. Soc.* **145**, 667–675 (2023)). Therefore, TCN was used as an extra moiety in an effort to locate the pillar[5]arene units in the frameworks. A single crystal of **MeP5-MOF-1** was immersed in a saturated TCN solution in acetone for 24 h, and the resulting single crystal was

picked for SCXRD data collection. Unfortunately, no diffraction was detected at high theta range, probably due to the random distribution of the guest molecules in the frameworks.

Tetrahydrofuran (THF) were also used as an extra moiety to locate the pillar[5]arene units in the frameworks by immersing **MeP5-MOF-1** crystals in THF at room temperature for 24 h, in which THF molecules could be visualized in the frameworks and pillar[5]arene cavities in **MeP5-MOF-5** (This work will be published in another paper). Here, **MeP5-MOF-5** was synthesized and confirmed with crystallographical order compared with **MeP5-MOF-1~MeP5-MOF-4**, where guest molecules like THF could be located in atomic resolution in the frameworks. Sadly, the resulting crystal structure of **MeP5-MOF-1** after THF solvent exchange (denoted as **MeP5-MOF-1-G**) did not allow for detection of the putative THF guests or resolution of the pillar[5]arene units on the struts were still not resolved. This facts reflect that THF molecules were unable to locate the pillar[5]arene units in **MeP5-MOF-1** in this case. The preparation of **MeP5-MOF-5**, resulting single crystal structures and SCXRD data of **MeP5-MOF-5** and **MeP5-MOF-1-G** are shown below:

Additional Fig. 4 Preparation of **MeP5-MOF-5**.

Additional Fig. 5 Capped-stick and spacefilling representation of the single crystal structure of $(\text{THF})_2@$ **MeP5-MOF-5**. Carbon atoms are grey, oxygen atoms are red, nitrogen atoms are blue, and zinc atoms are dark blue. Hydrogen atoms are omitted for clarity.

Additional Table 2 Experimental SCXRD data for $(\text{THF})_2@$ **MeP5-MOF-5**

Formula	$[\text{Zn}_2(\text{MeP5BPPy})(\text{En})] (\text{THF})_2$
Empirical formula	$\text{C}_{127}\text{H}_{106}\text{N}_2\text{O}_{18}\text{Zn}_2$
Formula weight	2078.87

Temperature/K	221
Crystal system	triclinic
Space group	$P\bar{1}$
$a/\text{\AA}$	16.8706(14)
$b/\text{\AA}$	18.7788(18)
$c/\text{\AA}$	20.7210(18)
$\alpha/^\circ$	102.446(4)
$\beta/^\circ$	95.740(4)
$\gamma/^\circ$	93.864(4)
Volume/ \AA^3	6351.4(10)
Z	2
$\rho_{\text{calc}}(\text{g}/\text{cm}^3)$	1.087
μ/mm^{-1}	0.611
$F(000)$	2172
Crystal size/ mm^3	$0.13 \times 0.12 \times 0.1$
Radiation	GaK α ($\lambda = 1.34139$)
2θ range for data collection/ $^\circ$	3.826 to 121.052
Index ranges	$-21 \leq h \leq 21, -23 \leq k \leq 24, -26 \leq l \leq 25$
Reflections collected	80876
Independent reflections	27794 [$R_{\text{int}} = 0.0789, R_{\text{sigma}} = 0.0956$]
Data/restraints/parameters	27794/748/1568
Goodness-of-fit on F^2	0.987
Final R indexes [$I \geq 2\sigma(I)$]	$R_1 = 0.0890, wR_2 = 0.2649$
Final R indexes [all data]	$R_1 = 0.1568, wR_2 = 0.3121$
Largest difference peak/hole/ $e \text{\AA}^{-3}$	0.82/−0.62
CCDC-number	unpublished

Supplementary Fig. 64 Capped-stick representation of the single crystal structure of MeP5-MOF-1-G. The pillar[5]arene units on the struts are disordered and not resolved. Carbon atoms are grey, oxygen atoms are red, nitrogen atoms are blue, and zinc atoms are dark blue. Hydrogen atoms are omitted for clarity.

Supplementary Fig. 65 Polyhedral representation of the single crystal structure of **MeP5-MOF-1-G** along the *c* axis (a) and the *b* axis (b). The elementary cell is marked with an orange cuboid ($a = 16.14 \text{ \AA}$, $b = 20.48 \text{ \AA}$, $c = 18.71 \text{ \AA}$). The pillar[5]arene units on the pillar struts are disordered and not resolved. Carbon atoms are grey, hydrogen atoms are white, oxygen atoms are red, nitrogen atoms are blue, and zinc atoms are dark blue.

Supplementary Table 16 Experimental SCXRD data of **MeP5-MOF-1-G**

Formula	$[\text{Zn}_2(\text{MeP5BPy})(\text{TPPE})]$
Empirical formula	$\text{C}_{35}\text{H}_{22}\text{NO}_4\text{Zn}$
Formula weight	585.9
Temperature/K	193
Crystal system	orthorhombic
Space group	Pmmm
$a/\text{\AA}$	16.143(3)
$b/\text{\AA}$	18.713(5)
$c/\text{\AA}$	20.483(3)
$\alpha/^\circ$	90
$\beta/^\circ$	90
$\gamma/^\circ$	90
Volume/ \AA^3	6187(2)
Z	2
$\rho_{\text{calc}}(\text{g}/\text{cm}^3)$	0.314
μ/mm^{-1}	0.35
$F(000)$	602
Crystal size/ mm^3	$0.13 \times 0.12 \times 0.1$
Radiation	$\text{CuK}\alpha$ ($\lambda = 1.54178$)
2θ range for data collection/ $^\circ$	4.722 to 136.472
Index ranges	$-11 \leq h \leq 19$, $-19 \leq k \leq 22$, $-20 \leq l \leq 24$

Reflections collected	34982
Independent reflections	6097 [$R_{\text{int}} = 0.1780$, $R_{\text{sigma}} = 0.1153$]
Data/restraints/parameters	6097/181/145
Goodness-of-fit on F^2	1.037
Final R indexes [$I \geq 2\sigma(I)$]	$R_1 = 0.1476$, $wR_2 = 0.3305$
Final R indexes [all data]	$R_1 = 0.1878$, $wR_2 = 0.3522$
Largest difference peak/hole/ $e \text{ \AA}^{-3}$	1.31/−1.10
CCDC-number	2267730

10. Reply to the tenth comment made by Reviewer 1 “The photographs in Figures 4a and b are not distinctly displaying the color differences among the crystals because of the varying background brightness levels in the two images.”

Many thanks. The photographs in Figures 4a and b were re-shot and the corresponding corrections have been made in the manuscript as shown below:

Fig. 4 | Supramolecular recognition studies of MeP5-MOF-1 and MeP5-MOF-2 with PQT and TCN. Optical microscopy images of MeP5-MOF-1: **a**, before uptake of PQT; **b**, after uptake of PQT; **c**, before uptake of TCN; **d**, after uptake of TCN. Scale bars, 200 μm . The mole ratios of PQT (**e**) and TCN (**f**) to struts in MeP5-MOF-1, MeP5-MOF-2 and Model-MOF-1, as inferred from ^1H NMR spectral studies of these MOFs after guest uptake in acetone. **g–i**, PXRD patterns of single crystalline samples of MeP5-MOF-1, MeP5-MOF-2 and Model-MOF-1: **I**, before guest uptake; **II**, after uptake of PQT; **III**, after uptake of TCN.

11. Reply to the eleventh comment made by Reviewer 1 “In addition to the fluorescence spectra of the PA unit, the authors should also demonstrate whether guest adsorption alters the fluorescence spectra of each MOF.”

Many thanks. Compared with the pillar[5]arene struts **MeP5BPy** and **MeP5BPPy**, the fluorescence emission of **MeP5-MOFs** should reflect the influence of the TPE moieties. Here, the emission ($\lambda = 450\text{--}500\text{ nm}$) of the pillar[5]arene-based struts overlaps with the emission ($\lambda = 450\text{--}550\text{ nm}$) of TPE moieties. The fluorescence spectra of **MeP5-MOFs** revealed a decrease after guest binding but still remained emissive notwithstanding the presence of the TPE moieties. An example is provided by a single crystal of **MeP5-MOF-1** under 365 nm UV light as shown below.

Additional Fig. 6 Optical microscopy image of a single crystal of **MeP5-MOF-1** under 365 nm UV light. Scale bars, 200 μm .

Supplementary Fig. 103 Solid-state fluorescence spectra of **MeP5-MOF-1** (~2 mg, black) before and after adding **PQT** (20.0 μM , red) or **TCN** (20.0 μM , blue) in acetone at room temperature, $\lambda_{\text{ex}} = 250\text{ nm}$.

Supplementary Fig. 104 Solid-state fluorescence spectra of **MeP5-MOF-2** (~2 mg, black) before and after adding **PQT** (20.0 μ M, red) or **TCN** (20.0 μ M, blue) in acetone at room temperature, $\lambda_{\text{ex}} = 250$ nm.

Supplementary Fig. 105 Solid-state fluorescence spectra of **MeP5-MOF-3** (~2 mg, black) before and after adding **PQT** (20.0 μ M, red) or **TCN** (20.0 μ M, blue) in acetone at room temperature, $\lambda_{\text{ex}} = 250$ nm.

Supplementary Fig. 106 Solid-state fluorescence spectra of **MeP5-MOF-4** (~2 mg, black) before and after adding **PQT** (20.0 μ M, red) or **TCN** (20.0 μ M, blue) in acetone at room temperature, $\lambda_{\text{ex}} = 250$ nm.

12. Reply to the twelfth comment made by Reviewer 1 “The author needs to clarify the toluene/pyridine separation condition of other studies to be compared. For example, in Ref. S14, 5.00 mg of adsorbent was added to 10 mL of 100:1 v/v Tol/Py mixture.”

Many thanks. The toluene/pyridine separation conditions of other studies as compared to those used in this work (the reference numbers have been changed to Refs. 17–19 from 14–16 in the Supplementary Information) are listed below:

In Ref. S16, 20.00 mg of adsorbent was added to 2 mL of a 50:50 v/v **Tol/Py** mixture.

In Ref. S17, 20.00 mg of adsorbent was added to 3 mL of a 50:50 v/v **Tol/Py** mixture.

In Ref. S18, 1.00 mg of adsorbent was added to 10 mL a of 100:100:1 v/v **Tol/Ben/Py** mixture. Here, **Ben** is short for benzene.

13. Reply to the thirteenth comment made by Reviewer 1 “Regarding the statement ‘We thus conclude that a judicious choice of receptor (e.g., pillar[5]arene) and MOF framework (e.g., MeP5-MOF-2) leads to the best observed Py removal and Py/Tol mixture separation capability compared with only pillar[5]arene or framework.’ on lines 299–301, the author needs to be cautious about this claim. The experimental conditions for the Tol/Py separation performance experiment in this work differ from those of the studies used for comparison by the authors.”

Many thanks. The corresponding corrections have been made in the manuscript. The statement has been changed as below: “We thus conclude that a judicious choice of receptor (e.g., pillar[5]arene) and MOF framework (e.g., **MeP5-MOF-2**) allows for the specific removal of **Py** from **Py/Tol** mixtures.” Here “the best observed” was replaced by “the specific” in this sentence.

14. Reply to the fourteenth comment made by Reviewer 1 “The author should provide data comparing the performance of their MOF with currently available commercial Tol/Py separation processes.”

Many thanks. Toluene (**Tol**) and benzene (**Ben**) are produced industrially by the process of coal coking and distillation but are often contaminated by a small amounts of pyridine (**Py**), which must be removed. However, not only do they have similar physical properties (boiling points, **Py**: 115.2°C, **Tol**: 110.6°C), toluene forms a minimum-boiling azeotrope with pyridine. This creates technical challenges for their separation and purification. Current methods to separate compounds that form minimum-boiling azeotropic mixtures include azeotropic distillation, extractive distillation, pressure-swing distillation, and others. Unfortunately, these methods suffer from disadvantages that include high cost, high energy consumption and technical complexity. Therefore, the removal of **Py** from **Tol** is still an unsolved problem (*Ind. Eng. Chem. Res.* **56**, 11894–11902 (2017)).

15. Reply to the fifteenth comment made by Reviewer 1 “The authors should provide additional data on the Tol/Py separation experiments, including information on the efficiency of the MOF adsorbents in bulk conditions, as well as their recyclability and stability in the Tol/Py separation process.”

Many thanks for the valuable advice. The efficiency of **MeP5-MOF-1** and **MeP5-MOF-2** under bulk separation conditions (~200 mg) was further investigated, and the recyclability and stability of the **Tol/Py** separation process is now discussed with these additional data in hand. **MeP5-MOF-1** and **MeP5-MOF-2** revealed similar results compared with the corresponding results of using ~20 mg MOF adsorbents. In these experiments, **MeP5-MOF-1** and **MeP5-MOF-2** were washed with acetone (10 mL) five times and used in next run. These cycles were conducted three times, and the efficiency was investigated by GC and ¹H NMR methods as shown below. The PXRD patterns revealed that **MeP5-MOF-1** and **MeP5-MOF-2** lost their partial crystallinity after the guest uptake.

Supplementary Fig. 148 GC measurements of the relative uptake of **Tol/Py** in **MeP5-MOF-1** after ~200 mg of crystals were placed in 1 mL of a 90:10 v/v **Tol/Py** mixture and allowed to stand for two minutes.

Supplementary Fig. 149 GC measurements of 1 mL of the residual 90:10 *v/v* (87.3:12.7 in mole percentage) **Tol/Py** mixture after a purification study involving adding ~200 mg crystals of **MeP5-MOF-1**. The mole percentage of **Tol** increased from 87.3% to 97.2% during the course of this experiment.

Supplementary Fig. 150 ^1H NMR spectra (500 MHz, $\text{DMSO-}d_6$: $\text{DCI} = 100:1$, 298 K): (a) initial **MeP5-MOF-1**; (b) initial **MeP5-MOF-1** after immersing in a 90:10 *v/v* **Tol/Py** mixture; (c) recycled **MeP5-MOF-1** washed with acetone (10 mL) five times. Under these conditions, about 70% guest molecules were removed; (d) recycled **MeP5-MOF-1** after immersing in a 90:10 *v/v* **Tol/Py** mixture.

Supplementary Fig. 151 GC measurements of the relative uptake of Tol/Py in MeP5-MOF-2 after ~200 mg of crystals were placed in 1 mL of a 90:10 v/v Tol/Py mixture and allowed to stand for two minutes.

Supplementary Fig. 152 GC measurements of 1 mL of the residual 90:10 v/v (87.3:12.7 in mole percentage) Tol/Py mixture after purification studies involving adding ~200 mg crystals of MeP5-MOF-2. The mole percentage of Tol increased from 87.3% to 90.6% during the course of this experiment.

Supplementary Fig. 153 ^1H NMR spectra (500 MHz, $\text{DMSO-}d_6$: $\text{DCI} = 100:1$, 298 K): (a) initial **MeP5-MOF-2**; (b) initial **MeP5-MOF-2** after immersing in a 90:10 v/v **Tol/Py** mixture; (c) recycled **MeP5-MOF-2** washed with acetone (10 mL) five times. Under these conditions, about 80% guest molecules were removed); (d) recycled **MeP5-MOF-2** after immersing in a 90:10 v/v **Tol/Py** mixture. The recycled selectivity of **Py** (mol%) based on the NMR spectra was calculated to be 90.2%.

Supplementary Fig. 154 PXRD patterns of **MeP5-MOF-1**: **I**, from a single crystal sample; **II**, after the sample was immersed in a 90:10 v/v **Tol/Py** mixture and washed with acetone (10 mL) five times; **III**, after the treatments in **II** were performed three cycles.

Supplementary Fig. 155 PXRD patterns of **MeP5-MOF-2**: **I**, from a single crystal sample; **II**, after the sample was immersed in a 90:10 *v/v* **Tol/Py** mixture and washed with acetone (10 mL) five times; **III**, after the treatments in **II** were performed three cycles.

16. Reply to the sixteenth comment made by Reviewer 1 “The authors should provide experimental data on the porosity of MeP5-MOFs, ideally through gas adsorption/desorption isotherm measurements.”

Many thanks. The porosity of **MeP5-MOF-1** and **MeP5-MOF-2** was investigated through CO₂ and N₂ adsorption/desorption isotherm measurements according to other related reports: (*Inorg. Chem.* **46**, 1233–1236 (2007); *Inorg. Chem.* **46**, 8490–8492 (2007)).

Supplementary Fig. 156 Experimental CO₂ adsorption/desorption isotherms at 195 K measuring the porosity of activated MeP5-MOF-1. The apparent BET surface area is calculated to be 160 m²/g.

Additional Fig. 7 Experimental N₂ adsorption/desorption isotherms at 77 K measuring the porosity of activated MeP5-MOF-1. The apparent BET surface area is calculated to be 5 m²/g, which is probably induced from the partial collapse of frameworks (*Inorg. Chem.* **46**, 1233–1236 (2007); *Inorg. Chem.* **46**, 8490–8492 (2007)).

Supplementary Fig. 157 PXRD patterns of MeP5-MOF-1: I, from a single crystal sample; II, after the sample was activated.

Supplementary Fig. 158 Experimental CO₂ adsorption/desorption isotherms at 195 K measuring the porosity of activated MeP5-MOF-2. The apparent BET surface area is calculated to be 190 m²/g.

Additional Fig. 8 Experimental N₂ adsorption isotherms at 77 K measuring the porosity of activated MeP5-MOF-2. The apparent BET surface area is calculated to be 3 m²/g, which is probably induced from the partial collapse of frameworks (*Inorg. Chem.* **46**, 1233–1236 (2007); *Inorg. Chem.* **46**, 8490–8492 (2007)).

Supplementary Fig. 159x PXRD patterns of MeP5-MOF-2: **I**, from a single crystal sample; **II**, after the sample was activated.

Reviewer #2's comments:

In this work, the authors present the synthesis of a series of MeP5-MOF-*n* single crystals based on pillar[5]arene-based ligands through a “pillar-layer” strategy. The introduction of macrocycle units into MOF frameworks exhibited intriguing guest recognition and separation properties. The efforts are impressive to grow and pick the single crystals and refine the SCXRD structure of complicated macrocycle-based MOFs with soft and disordered pillar[5]arene units. The current work is suitable for publication in *Nature Communication* after following minor revisions:

Response: We thank the reviewer for reading our manuscript so carefully and summarize the significance of our paper clearly. We also thank the reviewer for the positive comments. A point by point response is provided below:

1. Reply to the first comment made by Reviewer 2 “It is recommended that the authors revise the illustration figures for better clarity. In the current state, it is hard to distinguish different linkers of the same series (like MeP5BPY/MeP5BPPY and H₄TPE/H₄TPPE). I recommend using different colors for the same series of linkers (for example red for MeP5BPy and orange for MeP5BPPY, blue for H₄TPE and purple for H₄TPPE).”

Many thanks for the valuable advice. The corresponding corrections have been made in the manuscript in Figure 1. We have changed the colors for distinguishing different linkers for better clarity. The corresponding updated Figure 1 is shown in our response to Question 5 of Reviewer #1 above.

2. Reply to the second comment made by Reviewer 2 “In Figure S27, some C-C bonds on the benzene rings of ligands seems strange, showing square-shaped bonding. This Figure should be revised.”

Many thanks. The corresponding corrections have been made to the previous Figure S27 (the figure number has been changed to S29) in the Supplementary Information as shown below:

Supplementary Fig. 29 Polyhedral representation of the single crystal structure of **MeP5-MOF-1** along the *c* axis (a) and the *b* axis (b). The elementary cell is marked with an orange cuboid ($a = 16.45 \text{ \AA}$, $b = 20.20 \text{ \AA}$, $c = 18.31 \text{ \AA}$). The pillar[5]arene units on the pillar struts are disordered and not resolved. Carbon atoms are grey, hydrogen atoms are white, oxygen atoms are red, nitrogen atoms are blue, and zinc atoms are dark blue.

3. Reply to the third comment made by Reviewer 2 “SXR D indicated rigid framework backbone structures of the MeP5-MOF-*n* series. Is there any information about the permanent porosity of these MOFs from gas sorption analysis? How is the stability of these MOFs towards thermal treatment and ambient moisture?”

Many thanks. The porosity investigation has been carried out as shown in our response to Question 16 of Reviewer #1 above. The PXRD patterns revealed that **MeP5-MOF-1** and **MeP5-MOF-2** lost their partial crystallinity after gas adsorption.

Here, the zinc paddlewheel SBU is generally considered to be flexible, but still sufficiently robust not to undergo bond cleavage or rearrangement during desolvation of crystals (*J. Am. Chem. Soc.* **136**, 3776–3779 (2014)). The most conspicuous difference between **MeP5-MOF-1** and **MeP5-MOF-2** with and without solvent involves the coordination geometry of the bipyridine ligand: the (Zn···Zn)–N bonds are almost 180° in **MeP5-MOF-1** and **MeP5-MOF-2** with solvent, and the bonds become considerably bent in **MeP5-MOF-1** and **MeP5-MOF-2** without solvent. It appears that this angular distortion occurs in order to minimize the empty space in the desolvated structure, which also induces the changes of PXRD patterns.

The TGA experiments of **MeP5-MOF-1** and **MeP5-MOF-2** were performed. The resultant TGA curves revealed that **MeP5-MOF-1** had only a 3.6% weight loss before around 150°C which corresponded to the solvent loss and began to decompose at around 350°C while those of **MeP5-MOF-2** had a 3.9% weight loss before around 150°C and began to decompose at around 400°C .

Supplementary Fig. 160 TGA curve of the decomposition of **MeP5-MOF-1**.

Supplementary Fig. 161 TGA curve of the decomposition of **MeP5-MOF-2**.

The PXRD patterns as shown below revealed that **MeP5-MOF-1** and **MeP5-MOF-2** could maintain their partial crystallinity under solvated conditions. Additionally, PXRD patterns corresponding to solvent and vacuum treatment were also investigated and shown below in our response to Question 2 of Reviewer #3. The results confirmed that these MOFs were flexible and would lose their partial crystallinity under various conditions.

Supplementary Fig. 168 PXRD patterns of **MeP5-MOF-1**: **I**, from a single crystal sample; **II**, after the sample was immersed in deionized water at room temperature.

Supplementary Fig. 169 PXRD patterns of **MeP5-MOF-2**: **I**, from a single crystal sample; **II**, after the sample was immersed in deionized water at room temperature.

4. Reply to the fourth comment made by Reviewer 2 “MeP5-MOF-1 showed higher guest uptake than MeP5-MOF-2, what’s the possible reason?”

Many thanks. In this work, the structure of **MeP5-MOF-1** is non-interpenetrated and possesses open frameworks while that of **MeP5-MOF-2** is two-fold interpenetrated. The pore volume values of **MeP5-MOF-1** and **MeP5-MOF-2** were calculated according to the equation:

$$\text{Pore volume} = \frac{\text{Cell free volume}}{\text{Cell volume} \times \text{Density}}$$

The resultant values of pore volume are listed below:

Supplementary Table 25 The pore volumes of **MeP5-MOF-1** and **MeP5-MOF-2**.

Substance	Cell free volume (Å ³)	Density (g/cm ³)	Cell volume (Å ³)	Pore volume (cm ³ /g)
MeP5-MOF-1	4895.41	0.32	6084.43	2.52
MeP5-MOF-2	4884.7	0.75	8589.14	0.76

Therefore, we deduced that **MeP5-MOF-1** revealed a higher guest uptake than **MeP5-MOF-2** probably due to its higher pore volume.

5. Reply to the fifth comment made by Reviewer 2 “For Py and Tol separation, simulation and SCXRD refinement indicated that Py was most likely captured by the cavities of pillar[5]arene rings, while Tol was instead adsorbed between TPPE layers. Such distinct adsorption behaviors should be discussed in more detail from a chemical perspective to better understand the underlying guest separation mechanism.”

Many thanks. From the single crystal structure of **Py@P5**, **Py** binding is driven by [C–H···O] and [C–H··· π] interactions ([C···O] distances (Å), [H···O] distances (Å) and [C–H···O] angles (deg) of [C–H···O] hydrogen bonds: 3.42, 2.54, 154.88; 3.42, 2.54, 154.88. [C–H··· π] distances (Å) and angles (deg): 3.00, 158.34; 3.07, 158.76). These findings are consistent with the conclusion that **Py** can be captured by the pillar[5]arene cavities driven by noncovalent interactions.

According to the above analysis, we revised the corresponding statements of the guest separation mechanism in more detail in the manuscript as shown below:

“The resulting crystal structure revealed that one pillar[5]arene molecule can accommodate two **Py** molecules within its cavity (Fig. 5a). Based on the metric parameters, **Py** binding is driven by [C–H···O] and [C–H··· π] interactions ([C···O] distances (Å), [H···O] distances (Å) and [C–H···O] angles (deg) of [C–H···O] hydrogen bonds: 3.42, 2.54, 154.88; 3.42, 2.54, 154.88. [C–H··· π] distances (Å) and angles (deg): 3.00, 158.34; 3.07, 158.76. Supplementary Figs. 125–127).”

“Moreover, the **Tol** molecules are found within the voids between the **TPPE** layers rather than in the pillar[5]arene cavities (Supplementary Figs. 132–134). This stands in contrast to what is seen for **Py@P5**. We rationalize this difference in terms of the smaller size of the **Py** molecules which makes them more likely to be trapped in the pillar[5]arene cavities. We thus suggest that the pillar[5]arene units incorporated into **MeP5-MOF-2** endows the system with an ability to capture **Py** selectively from **Py/Tol** mixtures through a macrocycle-dependent recognition process and that this effect is enhanced by confinement within a framework (Figs. 5d–e).”

6. Reply to the sixth comment made by Reviewer 2 “What’s the guest sorption and separation properties of MeP5-MOF-3 and MeP5-MOF-4? No related properties were examined in the current manuscript.”

Many thanks. Compared with **MeP5-MOF-1** and **MeP5-MOF-2**, the preparation of **MeP5-MOF-3** and **MeP5-MOF-4** possess lower yields and worse crystal quality. The single crystals of **MeP5-MOF-3** and **MeP5-MOF-4** always exist polycrystals and have

cracks which makes them not suitable to be resolved by SCXRD. Therefore, the separation properties **MeP5-MOF-3** and **MeP5-MOF-4** have not been studied in the original work. Here, the selectivities for **Py** by **MeP5-MOF-3** and **MeP5-MOF-4** in the 90:10 v/v **Tol/Py** mixture were investigated. The results revealed that both **MeP5-MOF-3** and **MeP5-MOF-4** showed selectivities to **Py** but not as high as **MeP5-MOF-1** (90.3%) and **MeP5-MOF-2** (89.5%). Probably, the smaller size of TPE ligand in **MeP5-MOF-3** and **MeP5-MOF-4** prevented some guests from entering the frameworks and pillar[5]arene cavities, leading us to propose that **MeP5-MOF-1** and **MeP5-MOF-2** provide for better performance in removing **Py** from **Tol** as compared with **MeP5-MOF-3** and **MeP5-MOF-4**. Therefore, **MeP5-MOF-1** and **MeP5-MOF-2** were used to investigate the separation properties in detail.

Supplementary Fig. 146 GC measurements of the relative uptake of **Tol/Py** in **MeP5-MOF-3** after ~20 mg of crystals were placed in 100 μ L of a 90:10 v/v **Tol/Py** mixture and allowed to stand for two minutes.

Supplementary Fig. 147 GC measurements of the relative uptake of **Tol/Py** in **MeP5-MOF-4** after ~20 mg of crystals were placed in 100 μ L of a 90:10 v/v **Tol/Py** mixture and allowed to stand for two minutes.

7. Reply to the seventh comment made by Reviewer 2 “The determination of flexible groups in the structure through SCXRD may be more accurate at lower temperatures. Can SCXRD at lower temperatures (e.g., 80 K) be tested?”

Many thanks for this valuable advice. We have conducted the low temperature SCXRD experiments and the corresponding results are discussed in our response to Question 4 of Reviewer #1. Due to the worse quality of single crystals and greater disorder in the structures of **MeP5-MOF-3** and **MeP5-MOF-4**, **MeP5-MOF-1** was taken as an example to conduct the low temperature measurement. The SCXRD data of **MeP5-MOF-1** were collected at 105 K to investigate whether low temperature can indeed limit the dynamics of the pillar[5]arene units in the frameworks. From the single crystal structure, the pillar[5]arene units still could not be visually characterized even at 105 K. The single crystal structure analysis and the SCXRD data of **MeP5-MOF-1** collected at 105 K are presented in our response to Question 4 of Reviewer #1.

8. *Reply to the eighth comment made by Reviewer 2 “The ‘Mole Ratio’ in Fig. 4 should be added with error bar.”*

Many thanks. The corresponding corrections have been made in Fig. 4 and presented in our response to Question 10 of Reviewer #1.

9. *Reply to the ninth comment made by Reviewer 2 “The format of references should be noted, such as “macrocycle-based” in reference [16] and “clathrochelate-based” in reference [46].”*

Many thanks. The corresponding corrections have been done in the references of the manuscript as shown below: “Ji, X., Ahmed, M., Long, L., Khashab, N. M., Huang, F., Sessler, J. L. Adhesive supramolecular polymeric materials constructed from “macrocycle- based” host–guest interactions. *Chem. Soc. Rev.* **48**, 2682–2697 (2019); Gong, W., Xie, Y., Pham, T. D., Shetty, S., Son, F. A., Idrees, K. B., Chen, Z., Xie, H., Liu, Y., Snurr, R. Q., Chen, B., Alameddine, B., Cui, Y., Farha, O. K. Creating optimal pockets in a “clathrochelate- based” metal–organic framework for gas adsorption and separation: experimental and computational studies. *J. Am. Chem. Soc.* **144**, 3737–3745 (2022).”

Reviewer #3’s comments:

The authors reported the synthesis of pillararene-based MOFs and their use in host–guest recognition and separation of small molecules. Steric effect has been effective in hindering the rotation of linker pillars that resulted in the successful resolution of macrocyclic pillararenes within the MOF structure, which has been a challenge in previous related studies. The structural advance provides unequivocal evidence about the presence and location of macrocycles within the framework.

The pillararene-MOFs have shown improved and selective adsorption towards guests, and in one case being applied towards the separation of pyridine/toluene. Structural characterization was done thoroughly and thoughtfully. The authors however didn’t address the stability of the resulting MOFs, which raises substantial questions about their value towards host–guest binding and separation.

The manuscript is very well written overall, though I recommend the authors to address the following comments before its acceptance:

Response: We thank the reviewer for reading our manuscript so carefully and summarize the significance of our paper clearly. We also thank the reviewer for the positive comments. A point by point response is provided below:

1. Reply to the first comment made by Reviewer 3 “Porosity. For the interpenetrated MOF vs. the non-interpenetrated ones, are there significant differences in terms of surface area and pore size? Have the authors run BET studies of those MOFs?”

Many thanks. The porosity investigation has been done by BET studies and the corresponding results are shown in our response to Question 16 of Reviewer #1 above. Sorption of CO₂ at 195 K was performed since this kind of MOFs showed relative strong affinity to CO₂ compared to N₂ according to other related reports (*Inorg. Chem.* **46**, 1233–1236 (2007); *Inorg. Chem.* **46**, 8490–8492 (2007)). The apparent BET surface areas of **MeP5-MOF-1** and **MeP5-MOF-2** calculated by CO₂ are around 160 m²/g and 190 m²/g, respectively. The lower BET area of **MeP5-MOF-1** is probably caused by the structural collapse in the open frameworks. The PXRD patterns revealed that **MeP5-MOF-1** and **MeP5-MOF-2** lost their partial crystallinity after gas adsorption.

2. Reply to the second comment made by Reviewer 3 “The PXRD pattern of MOFs changed significantly before and after exposure to different guests. The authors attributed the changes to the dynamics within the MOF. This is quite handwaving. Can authors give more insight into such changes? This raises the question about the stability of the MOF towards solvents, vacuum, temperature etc. Are the host-guest responses reversible, i.e., can the PXRD be reverted after removal of guests? If not, what is the nature of the solid-state changes? It is reasonable to believe that the MOFs are assembled through the coordination between pyridyl groups of the pillars and the Zn metal centers, which is weak and may relate to the intrinsic instability of such MOFs. The authors are also suggested to run DSC to probe the thermal stability of the crystalline phases.”

Many thanks. Here, the changes in the PXRD patterns are ascribed to not only the dynamics of the frameworks and pillar[5]arene units but also the loss of partial crystallinity after each separation cycle.

It can be assumed that the (Zn···Zn)–N bond angle in solvent is almost linear while it becomes considerably bent in the desolvated state, and this will be also influenced by solvents. It appears that this angular distortion occurs in order to minimize the empty space in the desolvated structure, which can lead to the structural transformation and cause changes in the PXRD patterns (*J. Am. Chem. Soc.* **136**, 3776–3779 (2014)).

Moreover, the dynamics of pillar[5]arene units in the frameworks are also discussed above in our response to Questions 2, 3, 7 and 9 of Reviewer #1. The rotation of the pillar[5]arene units in the frameworks is a source of changes in the PXRD patterns.

We synthesized another pillar[5]arene-based MOF denoted as **MeP5-MOF-5** (this work will be published in another paper) with better crystallinity and crystallographical order compared with the MOFs in this work. The single crystal structures revealed that the structures could be resolved after exposure to different solvents such as tetrahydrofuran (THF), chlorobenzene (ClBz) and DMF. The solvent exchange process was performed by immersing the single crystals in the corresponding solvent and allowing to take up the guest molecules for 24 h at room temperature. The preparation of **MeP5-MOF-5**, single crystal structure and SCXRD data of (THF)₂@**MeP5-MOF-5** are shown in our response

to Question 9 of Reviewer #1. The single crystal structure and SCXRD data of (DMF)₆(ClBz)@MeP5-MOF-5 are shown below:

Additional Fig. 9 Capped-stick and spacefilling representation of the single crystal structure of (DMF)₆(ClBz)@MeP5-MOF-5. Carbon atoms are grey, oxygen atoms are red, nitrogen atoms are blue, zinc atoms are dark blue, and chlorine atom is green. Hydrogen atoms are omitted for clarity.

Additional Table 3 Experimental SCXRD data of (DMF)₆(ClBz)@MeP5-MOF-5

Formula	[Zn ₂ (MeP5BPPy)(En)] (DMF) ₆ (ClBz)
Empirical formula	C ₁₄₃ H ₁₃₇ ClN ₈ O ₂₂ Zn ₂
Formula weight	2485.79
Temperature/K	150
Crystal system	triclinic
Space group	P $\bar{1}$
a /Å	16.6344(7)
b /Å	19.2031(9)
c /Å	20.8426(10)
α /°	103.739(2)
β /°	94.507(2)
γ /°	93.572(2)
Volume/Å ³	6424.7(5)
Z	2
ρ_{calc} (g/cm ³)	1.285
μ /mm ⁻¹	0.467
F (000)	2608
Crystal size/mm ³	0.13 × 0.12 × 0.1
Radiation	MoK α (λ = 0.71073)
2 θ range for data collection/°	3.586 to 52.786
Index ranges	-20 ≤ h ≤ 20, -23 ≤ k ≤ 21, -26 ≤ l ≤ 26
Reflections collected	54935
Independent reflections	26134 [R _{int} = 0.0629, R _{sigma} = 0.1075]
Data/restraints/parameters	26134/48/1605
Goodness-of-fit on F ²	1.029

Final R indexes [$I \geq 2\sigma(I)$]	$R_1 = 0.0940, wR_2 = 0.2281$
Final R indexes [all data]	$R_1 = 0.1587, wR_2 = 0.2740$
Largest difference peak/hole/e \AA^{-3}	1.78/−0.81
CCDC-number	unpublished

The PXRD patterns of **MeP5-MOF-1** and **MeP5-MOF-2** after various solvent treatments were investigated as shown below:

Supplementary Fig. 164 Experimental PXRD patterns of **MeP5-MOF-1**: **I**, from a single crystal sample; **II**, after immersing in acetone; **III**, after immersing in ethanol; **IV**, after immersing in tetrahydrofuran. The various PXRD patterns after uptake of guests are considered reflective of the dynamics within the MOFs.

Supplementary Fig. 165 Experimental PXRD patterns of **MeP5-MOF-2**: **I**, from a single crystal sample; **II**, after immersing in acetone; **III**, after immersing in ethanol; **IV**,

after immersing in tetrahydrofuran. The various PXRD patterns after uptake of guests are considered reflective of the dynamics within the MOFs.

The PXRD patterns of **MeP5-MOF-1** and **MeP5-MOF-2** after desolvation under vacuum were investigated as shown below:

Supplementary Fig. 166 PXRD patterns of **MeP5-MOF-1**: **I**, from a single crystal sample; **II**, after the sample was desolvated under vacuum at 40 °C for 1 h.

Supplementary Fig. 167 PXRD patterns of **MeP5-MOF-2**: **I**, from a single crystal sample; **II**, after the sample was desolvated under vacuum at 40 °C for 1 h.

The TGA experiments of **MeP5-MOF-1** and **MeP5-MOF-2** were investigated. The corresponding results are shown in our response to Question 3 of Reviewer #2. The TGA curves revealed that **MeP5-MOF-1** had only a 3.6% weight loss before reaching about 150 °C, which corresponded to solvent loss. The sample began to decompose at around

350 °C. **MeP5-MOF-2** had a 3.9% weight loss before reaching roughly 150 °C and began to decompose at around 400 °C. DSC experiments were carried out. The corresponding results are shown below:

Supplementary Fig. 162 DSC analysis of **MeP5-MOF-1**. The broad peaks around 53 °C and 128 °C correspond to solvent loss.

Supplementary Fig. 163 DSC analysis of **MeP5-MOF-2**. The broad peaks around 46 °C and 167 °C correspond to solvent loss.

The host–guest responses were investigated through ^1H NMR methods and PXRD experiments as shown above in our response to Question 15 of Reviewer #1

(Supplementary Figs. 150, 153, 154 and 155). The host–guest responsibility experiments of **MeP5-MOF-1** and **MeP5-MOF-2** were investigated which revealed that some guest molecules were still trapped in **MeP5-MOF-1** and **MeP5-MOF-2** after each cycle according to the ^1H NMR results. The PXRD patterns revealed that **MeP5-MOF-1** and **MeP5-MOF-2** lost their partial crystallinity after guest uptake.

3. *Reply to the third comment made by Reviewer 3 “On a related note, as stated, better separation was achieved by repeated treatment of the Py/Tol mixture with fresh batches of MOF. Does the need of using fresh MOF also relate to the instability of the MOFs? The significance of their utility for separation will be greatly limited since it is of no practicality if the adsorbent can’t be recycled.”*

Many thanks. In this study, the separation could be repeated by washing **MeP5-MOF-1** and **MeP5-MOF-2** adsorbents with acetone five times, then recycled and used in next run. Fresh MOFs were used since trapped guest molecules in the used **MeP5-MOF-1** and **MeP5-MOF-2** could not be totally removed during the separation process. The digested samples were investigated by ^1H NMR and the results are presented in our response to Question 15 of Reviewer #1 (Supplementary Figs. 150 and 153).

Here, recycling experiments were conducted and the corresponding results are shown in our response to Question 15 of Reviewer #1 (Supplementary Figs. 150, 153, 154 and 155), in which the recyclability and stability of **MeP5-MOF-1** and **MeP5-MOF-2** in the Tol/Py separation process were discussed.

Reviewer #4’s comments:

In this manuscript, the authors reported a set of metal–organic frameworks (MOFs) containing incorporated pillar[5]arene units and used them in molecular recognition and selective separation. Since numerous macrocycle-incorporated MOFs cannot precisely locate the macrocycle moieties in the frameworks due to the rotation of macrocycles’ repeat units, the structure resolution of such MOFs is always a challenge. In this work, the authors altered the sizes of pillar[5]arene struts and obtained the atomically precise structures of pillar[5]arene-based MOFs characterized by single crystal X-ray diffraction. They found that the interpenetrated network limited the rotation of the pillar[5]arene repeat units in the frameworks. It is an interesting phenomenon, which is helpful for constructing macrocycle-incorporated crystalline frameworks in other cases. Their MOFs can recognize paraquat and 1,2,4,5-tetracyanobenzene in solution and selectively remove trace pyridine from toluene.

This manuscript is well written, and clearly illustrates the structure–property relationships with solid evidence. Therefore, I recommend this work to be accepted for publication after considering the following minor comments:

Response: We thank the reviewer for reading our manuscript so carefully and summarize the significance of our paper clearly. We also thank the reviewer for the positive comments. A point by point response is provided below:

1. *Reply to the first comment made by Reviewer 4 “Page 6, “After numerous attempts, single crystals of MeP5-MOF-2 were obtained utilizing a solvo-thermal procedure, strut MeP5BPPy was combined with H_4TPPE and $\text{Zn}(\text{NO}_3)_2 \cdot 6\text{H}_2\text{O}$.” Here, the authors should describe the conditions of single crystal growth with more details.”*

Many thanks. The corresponding corrections have been made in the manuscript as shown below: “Single crystals of **MeP5-MOF-2** were obtained *via* a solvo-thermal procedure, wherein strut **MeP5BPPy** was combined with 2 equiv of **H₄TPPE** and 2 equiv of Zn(NO₃)₂·6H₂O in DMF (Supplementary Fig. 9). Acetic acid was added as a modulator.”

2. Reply to the second comment made by Reviewer 4 “Page 7, “Single crystals of both species were obtained. The Flack parameters for *pS*-MeP5-MOF-2 and *pR*-MeP5-MOF-2 were found to be 0.28(2) and 0.26(3). These values are less than 0.30, allowing their absolute structures to be determined with confidence.” Here, the authors should cite the corresponding references about Flack parameters.”

Many thanks. The corresponding corrections have been made in the manuscript as shown below: “The Flack parameters for *pS*-MeP5-MOF-2 and *pR*-MeP5-MOF-2 were found to be 0.28(2) and 0.26(3). These values lead us to conclude that the absolute structures of these species could not be fully determined, probably due to the partial conformational interconversion between the *pS/pR*-MeP5 congeners during the preparation of *pS/pR*-MeP5-MOF-2 (Supplementary Tables 9–10)^{41–42}.” By way of calibration, the absolute structure of the crystal can be validated when the Flack parameter is below 0.1 with an enantiopure-sufficient inversion-distinguishing power. Key references regarding Flack parameters are cited in the manuscript as follows: *J. Appl. Cryst.* **33**, 1143–1148 (2000); *Angew. Chem. Int. Ed.* **25**, 11809–11813 (2021). We apologize for the misunderstanding of the absolute configuration determination with Flack parameter in the initial version of the manuscript and thank the referee for helping us avoid a scientific error.

3. Reply to the third comment made by Reviewer 4 “In this manuscript, the authors used TPE ligands as the layers to construct their pillar[5]arene-based MOFs. Were there any reasons to use the TPE ligands?”

Many thanks. The TPE materials are commercially available and easily acquired. This makes them suitable substrates for constructing the layers in this study. We also found that the TPE entities exhibited good solubility in DMF, and in some (but not all) cases allowed for the growth of single crystals suitable for structural analysis.

Having replied to all comments made by the four reviewers, we believe this manuscript is now ready for publication. In any case, we thank you for your kind consideration.

Best Regards,

Feihe

REVIEWERS' COMMENTS

Reviewer #1 (Remarks to the Author):

This revised version of the manuscript has responded well to the comments we previously provided and has significantly improved compared to the initially submitted paper. Therefore, we believe that this paper is now suitable for publication in Nature Communications.

Reviewer #2 (Remarks to the Author):

The authors have well addressed the questions, and therefore I recommend it for publication in Nature Communications.

Reviewer #3 (Remarks to the Author):

The authors have done a thorough job in responding to all the comments. I recommend its acceptance but for the completeness, I would suggest the authors to address the following:

1. Since the experimental data convincingly confirmed the instability of the crystal structures of most of the MOFs towards various environmental conditions, it is necessary, in my opinion, to clearly state this behavior in the main text. The authors should also mention the BET results in the main text. For readers who care, it would give them a sense of what is most relevant when designing host-containing MOFs for better separation. It appears to me that the exact crystal structure of the MOF isn't important, nor is the porosity.

Reviewer #4 (Remarks to the Author):

The revised manuscript is recommended for publication.

RESPONSE TO REVIEWERS' COMMENTS

We very appreciate the valuable comments made by the four reviewers. They were very helpful and allowed us to improve our manuscript. We recognize the wisdom of the reviewers and have revised the manuscript and supplementary information as requested by the reviewers. The corrections and improvements are listed below in a point by point fashion in response to the reviewers' comments.

Reviewer #1's comments:

This revised version of the manuscript has responded well to the comments we previously provided and has significantly improved compared to the initially submitted paper. Therefore, we believe that this paper is now suitable for publication in *Nature Communications*.

Response: We thank the reviewer for providing a positive comment and allowing publication of this paper in *Nature Communications*.

Reviewer #2's comments:

The authors have well addressed the questions, and therefore I recommend it for publication in *Nature Communications*.

Response: We thank the reviewer for providing a positive comment and allowing publication of this paper in *Nature Communications*.

Reviewer #3's comments:

The authors have done a thorough job in responding to all the comments. I recommend its acceptance but for the completeness, I would suggest the authors to address the following: 1. Since the experimental data convincingly confirmed the instability of the crystal structures of most of the MOFs towards various environmental conditions, it is necessary, in my opinion, to clearly state this behavior in the main text. The authors should also mention the BET results in the main text. For readers who care, it would give them a sense of what is most relevant when designing host-containing MOFs for better separation. It appears to me that the exact crystal structure of the MOF isn't important, nor is the porosity.

Response: We thank the reviewer for providing a positive comment and allowing

publication of this paper in *Nature Communications*.

Moreover, we thank the reviewer's advice and have added the discussion of stability and BET results of the MOFs in the main text as shown below:

“We further investigated the efficiency of both **MeP5-MOF-1** and **MeP5-MOF-2** under bulk conditions (~200 mg). Both **MeP5-MOF-1** and **MeP5-MOF-2** gave similar results with samples (~20 mg for each) tested as adsorbents (Supplementary Figs. 148–153). The recyclability of the separation process is discussed in these cases. The host–guest responsibility experiments of **MeP5-MOF-1** and **MeP5-MOF-2** revealed that some guest molecules were still trapped in **MeP5-MOF-1** and **MeP5-MOF-2** after each cycle according to the ¹H NMR results. It is worth noting that the PXRD patterns reflected **MeP5-MOF-1** and **MeP5-MOF-2** lost their partial crystallinity after each cycle of the guest uptake (Supplementary Figs. 154–155). We also investigated the environmental tolerance of **MeP5-MOF-1** and **MeP5-MOF-2** under various treatment. The porosity of **MeP5-MOF-1** and **MeP5-MOF-2** was studied through CO₂ and N₂ adsorption/desorption measurements (Supplementary Figs. 156–161). Experimental CO₂ adsorption/desorption isotherms at 195 K measuring the porosity of activated **MeP5-MOF-1** and **MeP5-MOF-2** revealed that the apparent Brunauer-Emmett-Teller (BET) surface areas are calculated to be 160 m²/g and 190 m²/g, respectively. These MOFs are non-porous to N₂ as revealed by N₂ sorption experiments at 77 K. This difference toward CO₂ and N₂ could be ascribed to that these MOFs showed relative strong affinity to CO₂ compared to N₂ according to other related report.⁴⁹ The TGA experiments of **MeP5-MOF-1** and **MeP5-MOF-2** were performed to investigate their decomposing temperatures. The resultant TGA curves revealed that **MeP5-MOF-1** had only a 3.6% weight loss before around 150 °C which corresponded to the solvent loss and began to decompose at around 350 °C while those of **MeP5-MOF-2** had a 3.9% weight loss before around 150 °C and began to decompose at around 400 °C (Supplementary Figs. 162–165). After treatments in some specific solvents, the PXRD patterns revealed that **MeP5-MOF-1** and **MeP5-MOF-2** could still maintain their crystallinity (Supplementary Figs. 166–167). Additionally, PXRD patterns corresponding to wet and vacuum treatment were also investigated and confirmed again that these MOFs were dynamic and would lose their partial crystallinity under some specific conditions (Supplementary Figs. 168–171).”

Reviewer #4's comments:

The revised manuscript is recommended for publication.

Response: We thank the reviewer for providing a positive comment and allowing publication of this paper in *Nature Communications*.

Having replied to all comments made by the four reviewers, we believe this manuscript is now ready for publication. In any case, we thank you for your kind consideration.

Best Regards,

Feihe